# Dissecting the human leptomeninges at single-cell resolution

Nicola A. Kearns[1,6], Artemis Iatrou[1,4,6], Daniel J. Flood[1], Sashini De Tissera[1], Zachary M. Mullaney[1], Jishu Xu[1], Chris Gaiteri[1,5], David A. Bennett[1] & Yanling Wang [ID][1,2,3] ✉

Emerging evidence shows that the meninges conduct essential immune surveillance and immune defense at the brain border, and the dysfunction of meningeal immunity contributes to aging and neurodegeneration. However, no study exists on the molecular properties of cell types within human leptomeninges. Here, we provide single nuclei profiling of dissected postmortem leptomeninges from aged individuals. We detect diverse cell types, including unique meningeal endothelial, mural, and fibroblast subtypes. For immune cells, we show that most T cells express CD8 and bear characteristics of tissue-resident memory T cells. We also identify distinct subtypes of border-associated macrophages (BAMs) that display differential gene expressions from microglia and express risk genes for Alzheimer's Disease (AD), as nominated by genome-wide association studies (GWAS). We discover cell-type-specific differentially expressed genes in individuals with Alzheimer's dementia, particularly in fibroblasts and BAMs. Indeed, when cultured, leptomeningeal cells display the signature of ex vivo AD fibroblasts upon amyloid-β treatment. We further explore ligand-receptor interactions within the leptomeningeal niche and computationally infer intercellular communications in AD. Thus, our study establishes a molecular map of human leptomeningeal cell types, providing significant insight into the border immune and fibrotic responses in AD.

The meninges are a three-layered fibrous covering of the central nervous system (CNS), comprised of the pia, arachnoid (collectively known as the leptomeninges), and dura. The subarachnoid space (SAS), formed between the pia and arachnoid barrier layer, is filled with cerebrospinal fluid (CSF) and contains blood vessels and immune cells. The arachnoid trabeculae, spanning the SAS, bridge the arachnoid barrier layer and the fibroblast-rich pia mater. Because of tight junctions between the arachnoid barrier cells, the arachnoid barrier layer creates a blood-CSF (BCSF) barrier between the fenestrated vasculature of the dura and SAS compartment. Like the parenchymal blood-brain barrier (BBB), the leptomeningeal vasculature forms a specialized blood-meningeal barrier (BMB) that exhibits selective impermeability to immune cells and most macromolecules.

Although accumulated studies on brain parenchyma have shown that the vascular stability, BBB function, and cell immunity are substantially declined in aging and neurodegeneration[1–4], border tissues such as meninges, despite its extensive vascular network, a repertoire of immune cells and direct exposure to CSF-borne antigens, are understudied. Emerging evidence shows that the meninges act as a functional neuro-immune interface to maintain brain homeostasis in

[1]Rush Alzheimer's Disease Center, Rush University Medical Center, Chicago, IL 60612, USA. [2]Department of Neurological Sciences, Rush University Medical Center, Chicago, IL 60612, USA. [3]Rush Graduate College, Rush University, Chicago IL 60612, USA. [4]Present address: Department of Psychiatry, McLean Hospital, Harvard Medical School, Belmont, MA 02478, USA. [5]Present address: Department of Psychiatry, Upstate Medical University, Syracuse, NY 13210, USA. [6]These authors contributed equally: Nicola A. Kearns, Artemis Iatrou. ✉e-mail: yanling_wang@rush.edu

health and disease[5]. The immune surveillance and immune defense of the meninges, i.e., meningeal immunity, significantly influences spatial learning, memory, and social behaviors in animal models[1,2,6–9]. Accordingly, disruption of meningeal immunity promotes amyloid-β deposition and exacerbates the microglial inflammatory response in transgenic mouse models of Alzheimer's disease (AD)[6,10]. Importantly, immune cells within meninges actively interact with other cell types to conduct immune surveillance[7], suggesting signaling crosstalk between meningeal immune cells and neighboring cells.

Recent single-cell/nuclei RNA sequencing (sc/snRNA-seq) studies have begun to reveal the transcriptional diversity of meningeal cells. In developing mouse meninges, fibroblasts from three meningeal layers demonstrate distinct molecular signatures[11]. The adult mouse brain meninges contain a rich transcriptional diversity of immune cells, especially region-specific border-associated macrophages (BAMs)[12]. A recent scRNA-seq study of dura reveals diverse and heterogeneous immune and non-immune cell types in meningioma and surrounding non-tumor dura[13]. However, no single-nuclei RNA-seq study on dissected human leptomeninges has been conducted to date. Therefore, there is a critical need to unbiasedly chart cell types and states of human leptomeninges in normal aging and disease.

We address this challenge by reporting a comprehensive single-nuclei characterization of 42,557 cells from isolated postmortem human leptomeninges of aged individuals. Our study provides a transcriptomic atlas of the aged human leptomeninges, revealing rich cell type diversity within the stromal and immune cell types. We also investigate cell-type-specific gene expression changes and intercellular communications in AD leptomeninges.

## Results

### snRNA-seq reveals major cell types of the human leptomeninges

To transcriptionally characterize the leptomeningeal cell types, we performed snRNA-seq on 18 aged individuals (NCI/MCI = 9, AD = 9; Supplementary Data 1). We employed a modified VINE-seq[14] protocol to facilitate vascular nuclei extraction and recovered 46,121 total nuclei after single nuclei capture and sequencing. We then integrated nuclei across individuals using harmony[15], clustered them with the Louvain algorithm, and visualized cell clusters using uniform manifold approximation and projection (UMAP). Based on canonical markers, we annotated cell clusters and detected four major leptomeningeal cell types, including endothelial, mural, fibroblast, and immune cell populations (Supplementary Fig. 1a–c, Supplementary Data 2). We also detected small numbers of neurons, astrocytes, oligodendrocytes, and microglia (Supplementary Fig. 1a–c) from the underlying parenchyma due to its tight association with the pia mater. Gene detection per cell was comparable to other snRNA-seq studies on postmortem human nuclei[14,16,17], with no significant cell-subtype proportional differences between NCI/MCI and AD groups (Supplementary Fig. 1d, e; Supplementary Data 3). Next, we excluded five small parenchymal cell clusters (3564 nuclei) and re-clustered the remaining 42,557 nuclei to form a leptomeningeal dataset. Hierarchical clustering revealed two subtypes within each major cell type, and each subtype presented distinct gene expression profiles (Fig. 1b, c). To confirm the presence of each major cell type within the leptomeninges, we performed H&E staining and IHC on consecutive sections. H&E staining showed that leptomeninges harbored scattered cells, numerous large and small vessels, and an extensive extracellular matrix (ECM) (Fig. 1d). We then validated the major cell types by IHC staining, including PECAM+ endothelial cells, ACTA2+ mural cells, DCN+ fibroblasts, and CD45+ immune cells (Fig. 1d). Notably, fibroblasts and immune cells were observed in the ECM-rich sub-arachnoid spaces between vessels. In summary, our results reveal transcriptionally diverse endothelial, mural, immune, and fibroblast cells within the aged human leptomeninges.

### Joint analysis of leptomeningeal and parenchymal vascular cells recapitulates a complete continuum of the arteriovenous axis

Leptomeningeal vessels form an extensive network that penetrates the underlying cortex. The anatomical structure of blood vessels changes as leptomeningeal arteries progress to parenchymal arterioles, capillaries, venules, and back to leptomeningeal veins along the anatomical arteriovenous axis[18,19]. We reasoned that integrating the leptomeningeal dataset with recent snRNA-seq data of cortical vessels from Yang et al[14]. would enable us to gain a more comprehensive view of endothelial and mural cell types of human CNS vessels. To this end, we first integrated the endothelial nuclei from our study with those from Yang et al. and performed joint cell clustering. Our analysis revealed subclusters of arterial (aEndo), capillary (capEndo), and venous (vEndo) endothelial cells (Fig. 2a, b). Cell proportion analysis showed opposing compositions between the two studies, with aEndo and vEndo enriched in the leptomeninges and capEndo enriched in the frontal cortex and hippocampus (Fig. 2c), mirroring the anatomical vessel structure differences between leptomeninges and cortical parenchyma. To further characterize the transcriptional profiles of the aEndo, vEndo_1, and vEndo_2 populations of the leptomeninges, we performed differential gene expression analysis and identified markers that distinguish those three major endothelial clusters (Supplementary Fig. 2a–c, Supplementary Data 4).

Next, we analyzed the joint mural dataset and revealed a total of 5 clusters: two arterial/arteriole smooth muscle cell (aSMC) clusters, two pericyte subclusters, and a venous SMC cluster (vSMC; Fig. 2d, e). Notably, mural cells also showed opposing representations of cell types between the two studies, with SMCs enriched in the leptomeningeal dataset and pericytes enriched in the cortical dataset (Fig. 2d–f).To further characterize the transcriptional profiles of the aSMC_1, aSMC_2, and vSMC populations, we performed differential gene expression analysis and identified markers that distinguish those three major mural cell clusters (Supplementary Fig. 2d, e, Supplementary Data 4).

The vSMC cluster was only detected in the leptomeningeal dataset, indicating distinctive characteristics of leptomeningeal veins. To further validate the presence of vSMC, we leveraged a recently published study that profiled arterial and venous SMCs across murine tissues[20]. In line with this study, we found that *ACTA2* was expressed across all three leptomeningeal SMC clusters, but *CSPG4* expression was restricted to the aSMC clusters (Fig. 2g). We further confirmed this finding by immunohistochemistry. We detected CSPG4 only on leptomeningeal arteries, whereas ACTA2+ SMCs were found surrounding all vessels (Fig. 2g). Next, we generated a murine vSMC gene module based on Muhl et al.[20]. and examined this module's expression among the leptomeningeal SMC clusters. Indeed, we found that expression of the module was enriched only in the leptomeningeal vSMC cluster but not in leptomeningeal aSMC clusters (Fig. 2h). We then conducted differential gene expression analysis between the leptomeningeal aSMC and vSMCs. Notably, we detected increased expression of *APOE* and *LPL* in vSMCs, suggesting a potential role for the vSMCs in amyloid beta aggregation and clearance (Fig. 2i).

Because leptomeningeal vasculature is the primary site of entering and exiting the brain parenchyma, we reasoned that we could order endothelial and mural nuclei with the joint dataset to construct the zonation continuum of the vasculature in the compartment of leptomeninges and parenchyma. Indeed, the pseudotime trajectory (Figs. 2j, k) recapitulated a complete continuum of anatomical arteriovenous axis from leptomeningeal arteries to parenchymal arterioles, capillaries, venules and back to leptomeningeal veins (Fig. 2l). By clustering the significant trajectory-variable genes (*n* = 515 for endothelial cells; *n* = 302 for mural cells), we detected six gradient patterns of gene expression for endothelial and mural cells along the arteriovenous axis (Fig. 2j, k; Supplementary Data 4).

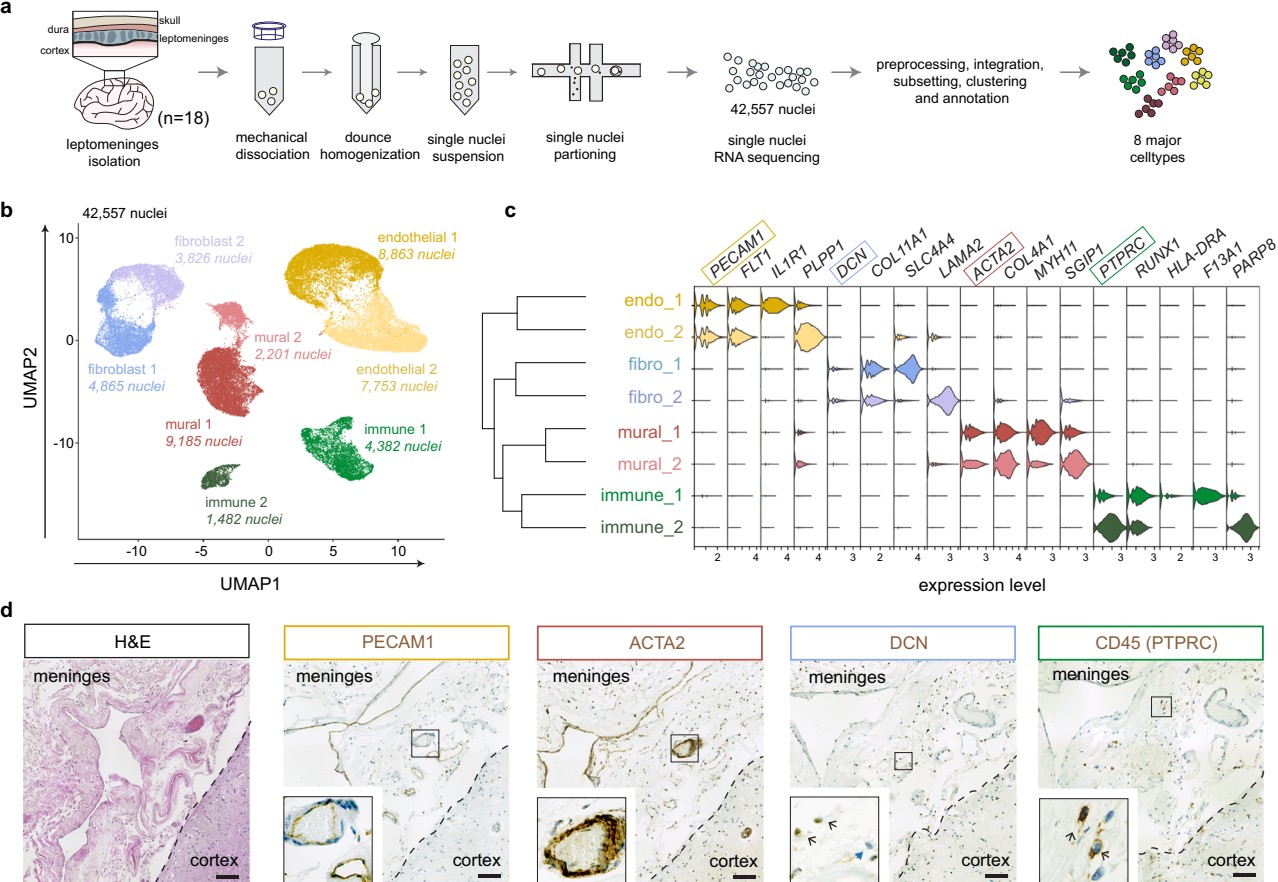

**Fig. 1 | Single nuclei sequencing of the human leptomeninges. a** Schematic of study design and single nuclei isolation protocol. **b** UMAP of meningeal cell types integrated across all donors, colored by coarse cell type clusters. **c** Hierarchical clustering of cell type clusters and violin plot of canonical and cluster-specific genes. **d** Representative H&E and chromogenic immunohistochemistry of PECAM1 (endothelial) ACTA2 (mural), DCN (fibroblast), and CD45 (immune) proteins from consecutive FFPE postmortem human leptomeninges sections, *n* = 3 individuals. Scale bars: 100 μm.

## Leptomeningeal fibroblast subtypes and identification of arachnoid barrier-like cells

We then characterized the fibroblast cell types from isolated human leptomeninges by further sub-clustering the fibroblast population from our snRNA-seq dataset. Interestingly, we identified four transcriptionally distinct subtypes (Fig. 3a, b; Supplementary Data 5): the first two subclusters, Fibro_1a and Fibro_1b, were characterized by high expression of membrane solute carrier transporter genes *SLC4A4* and *SLC7A2*; the remaining two subclusters, Fibro_2a and Fibro_2b, by high expression of *LAMA2* (Fig. 3b, e). We then created a network of GO terms derived from the top marker genes from each fibroblast subtype. Overall, our analysis suggested that the fibroblast subtypes in the leptomeninges were functionally associated with vascular transport, cell-matrix adhesion, ECM organization, and cellular communications (Fig. 3c).

To validate fibroblast subtypes, we tested a set of probes against top subtype markers, *SLC7A2* for Fibro_1a and Fibro_1b, *LAMA2* for Fibro_2a and Fibro_2b, and *TPRM3* for Fibro_1b and Fibro_2a (Fig. 3d). We then quantified spot signals per cell for each probe (*n* = 951 cells from 3 individuals) and analyzed the co-expression between probes. We confirmed those four major cell types (Fig. 3d, e), and their proportions were comparable to those quantified by snRNA-seq (Fig. 3f). We also detected ~7% of cells expressing all three markers, which could be due to the relatively high sensitivity of RNAscope for selected probes.

Recent snRNA-seq studies have reported the transcriptional signatures of brain fibroblasts, but those data were generated from experimentally isolating parenchymal vasculatures[14] or in silico sorting of human parenchymal nuclei[16]. Therefore, whether fibroblasts detected by those studies resemble the fibroblasts residing within the leptomeninges compartment remains unknown. To address this, we integrated our leptomeninges dataset with the frontal cortex and the hippocampal datasets from Yang et al.[14]. Joint cell clustering revealed that the perivascular and meningeal fibroblasts reported by Yang et al. transcriptionally most resembled leptomeningeal Fibro_2a and Fibro_1a, respectively (Supplementary Fig. 3a–c). Notably, Fibro_1b was only detected in the leptomeninges (Supplementary Fig. 3a–c) and expressed tight junction protein markers *CLDN11* and *TJP1*, the arachnoid marker *PTDGS*[11], and the arachnoid barrier markers *VIM* and *CDH1*[21] (Fig. 3g), suggesting the arachnoid barrier cell identity of Fibro_1b cluster. To validate it, we performed RNAscope analysis on the arachnoid barrier region (Fig. 3h, i). Indeed, we found that *SLC7A2* + *TPRM3* + Fibro_1b cells were enriched in the arachnoid barrier layer (42%) relative to the entire leptomeninges (18%) (Fig. 3j, k, 3f).

## Immune cell diversity in the human leptomeninges

As the leptomeninges is a key site of immune surveillance of the brain, we next characterized the diversity of immune cells within the leptomeninges. Sub-clustering of the meningeal immune clusters revealed four major subtypes, including BAMs, monocytes, T cells, and B cells (Fig. 4a, b). Notably, the BAM markers *LYVE1* and *CD163* have previously been reported to distinguish leptomeningeal BAMs in mice[12,22]. We then conducted immunohistochemistry with antibodies against CD3 for T cells and F13A1 for BAMs and detected both immune cell types within the leptomeninges (Fig. 4c, h).

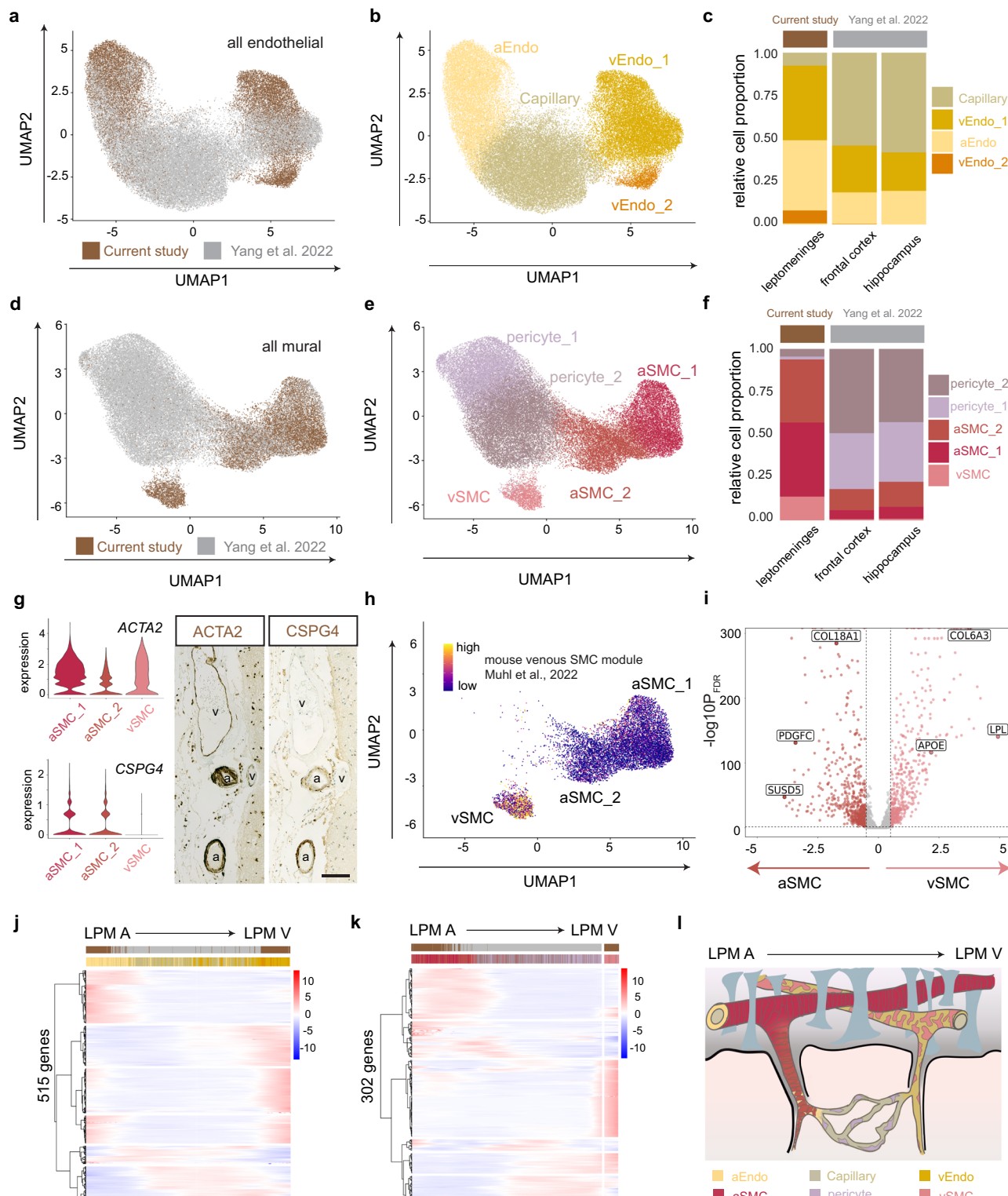

**Fig. 2 | Joint analysis of leptomeningeal and parenchymal vascular cells reca-pitulates a complete continuum of the pia-parenchyma-pia arteriovenous axis.**
**a–f** UMAP of integrated endothelial and mural subtypes from leptomeninges (current study) and parenchyma[14] colored by study (**a**, **d**) or cell type (**b**, **e**). Pro-portional differences of each subtype across each brain region are presented on the right (**c**, **f**). **g** Violin plots of ACTA2 and CSPG4 expression across the smooth muscle cell (SMC) types identified in the current study and representative chromogenic immunohistochemistry, $n = 3$ individuals. Scale bars:100 μm. a: artery, v- vein.

**h** Enrichment of a mouse venous SMC gene module[20] in the SMC cells from the current study. **i** Volcano plot of the differentially expressed genes distinguishing arterial from venous SMCs using a negative binomial generalized mixed model, Bonferroni correction; |logFC| > 0.5 and pBON <0.01 are colored. **j**, **k** Heatmap of zonation-dependent gene expression from integrated endothelial (b) and mural (**k**) cells. **l** Anatomical reference of the pia-parenchyma-pia arteriovenous continuum. LPM A: leptomeningeal arteries; LPM V: leptomeningeal veins.

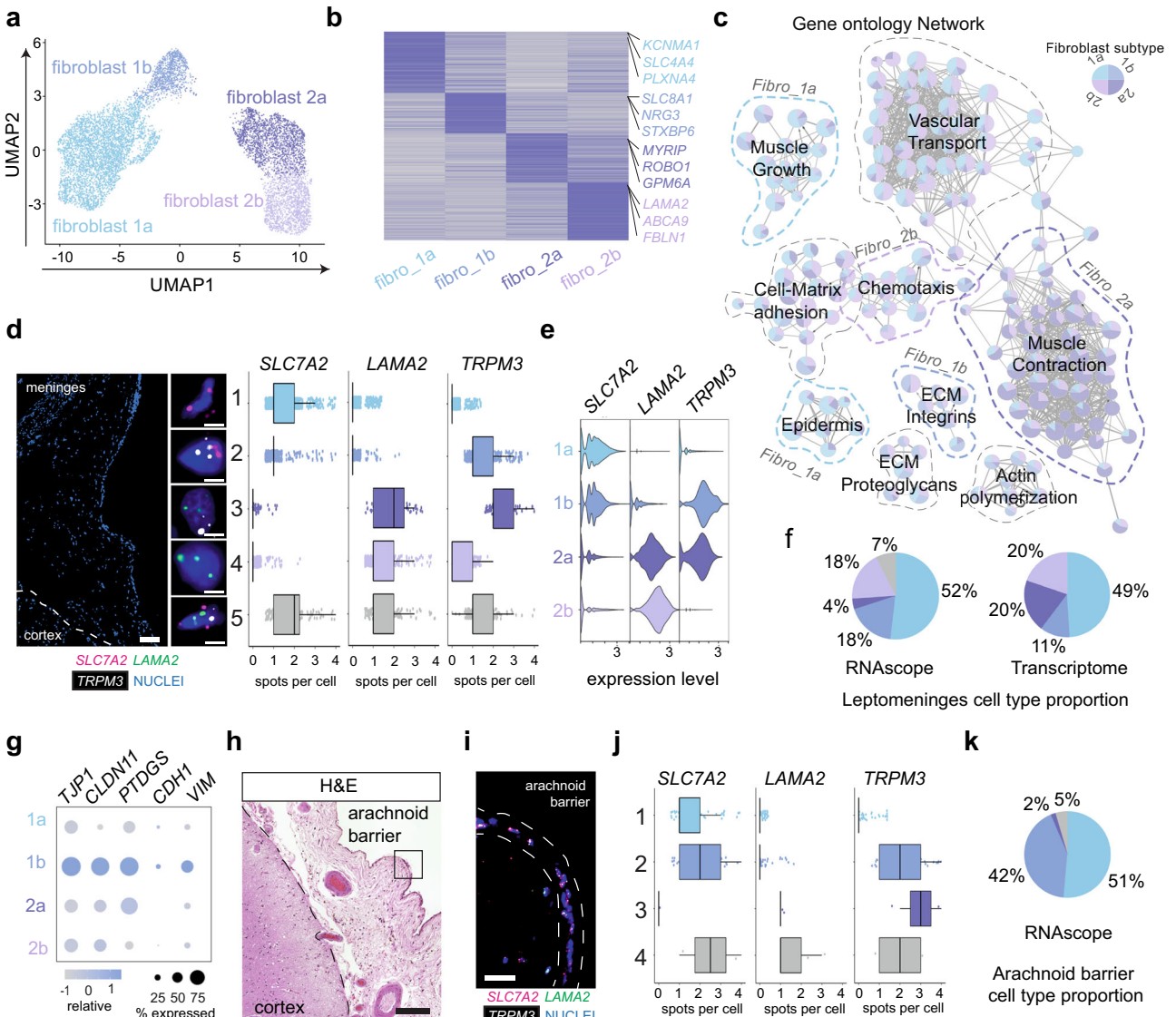

**Fig. 3 | Leptomeningeal fibroblast subtypes and identification of arachnoid barrier-like cells. a** UMAP of leptomeningeal fibroblast cells. **b** Heatmap of the top 100 markers for each fibroblast subtype, with representative genes annotated on the right. **c** Gene ontology network of shared and enriched pathways in fibroblast subtypes. **d** A representative RNAscope image of human leptomeninges showing transcripts of *SLC7A2* (fibro_1a and 1b), *LAMA2* (fibro_2a and 2b), and *TRPM3* (fibro_1b and 2a). Boxplots of transcript spot quantification of each of five cell types based on probe expression, *n* = 951 cells from 3 individuals. All points are plotted with overlaid boxplots, center lines show the medians; box limits indicate the 25th and 75th percentiles; whiskers extend 1.5 times the interquartile range from the 25th and 75th percentiles. Scale bars = 100 μm (overview), 10 μm (insets). **e** Violin plot of *SLC7A2*, *LAMA2*, and *TPRM3* expression by fibroblast subtype from the

snRNA-seq analysis. **f** Pie charts of fibroblast subtype proportions as quantified by RNAscope and snRNA-seq analysis. **g** Dot plot of *TJP1, CLDN11, PTGDS, CDH1, and VIM* expression by fibroblast subtype from the snRNA-seq data. **h** A representative H&E image of the leptomeninges showing the arachnoid barrier region, *n* = 3 individuals (**i**) A representative RNAscope image of the arachnoid barrier region showing transcripts of *SLC7A2*, *LAMA2*, and *TRPM3*. Scale bar = 20 μm. **j** Four cell types are identified based on probe expression pattern, and the boxplots show transcript spot quantification of each cell type, *n* = x cells. All points are plotted with overlaid boxplots, center lines show the medians; box limits indicate the 25th and 75th percentiles; whiskers extend 1.5 times the interquartile range from the 25th and 75th percentiles. **k** Pie charts of fibroblast subtype proportions in the arachnoid barrier region.

To examine whether the leptomeningeal niche affects T cell states and functional response, we performed iterative clustering of the T cells in our dataset and resolved them into 6 clusters, including four *CD8*+ clusters, one *CD4*+ cluster, and one cluster negative for both *CD4* and *CD8* (Fig. 4d, e). Interestingly, all cell clusters expressed core genes of tissue-resident memory T cells (TRMs), including *CXCR4, CD44, CD69,* and *ITGAE*[23] (Fig. 4f). The *CD8*+ clusters 1–4 expressed *CAMK4, USP36, HSP90AA,* and *SGCD,* respectively. Cluster 5 expressed high levels of *NCAM*1 and *KLRF1,* classical markers for NK/NKT cells. *CD4*+ cluster 6 expressed *LEF1* and *PVT1* (Fig. 4g, Supplementary Data 6). Together, these data showed that the leptomeninges harbor a repertoire of transcriptionally distinct T cells, suggesting the

leptomeningeal local niche may shape T cell-specific states for local immune surveillance and antigen recognition.

Recent mouse studies have shown that meningeal BAMs and microglia share the same embryonic origin and core gene signature, albeit some genes are highly restricted to either microglia or BAMs[12,24]. To compare meningeal BAMs with parenchymal microglia in humans, we took advantage of the microglial cluster detected in our dataset and performed differential gene expression analysis. We detected 780 DEGs, of which 402 genes were highly restricted to BAMs, and 378 genes to microglia (Fig. 4i; Supplementary Data 7). In line with the previous mouse scRNA-seq study[12], we also detected genes such as *LYVE1* and *COLEC12* were specifically expressed in meningeal BAMs,

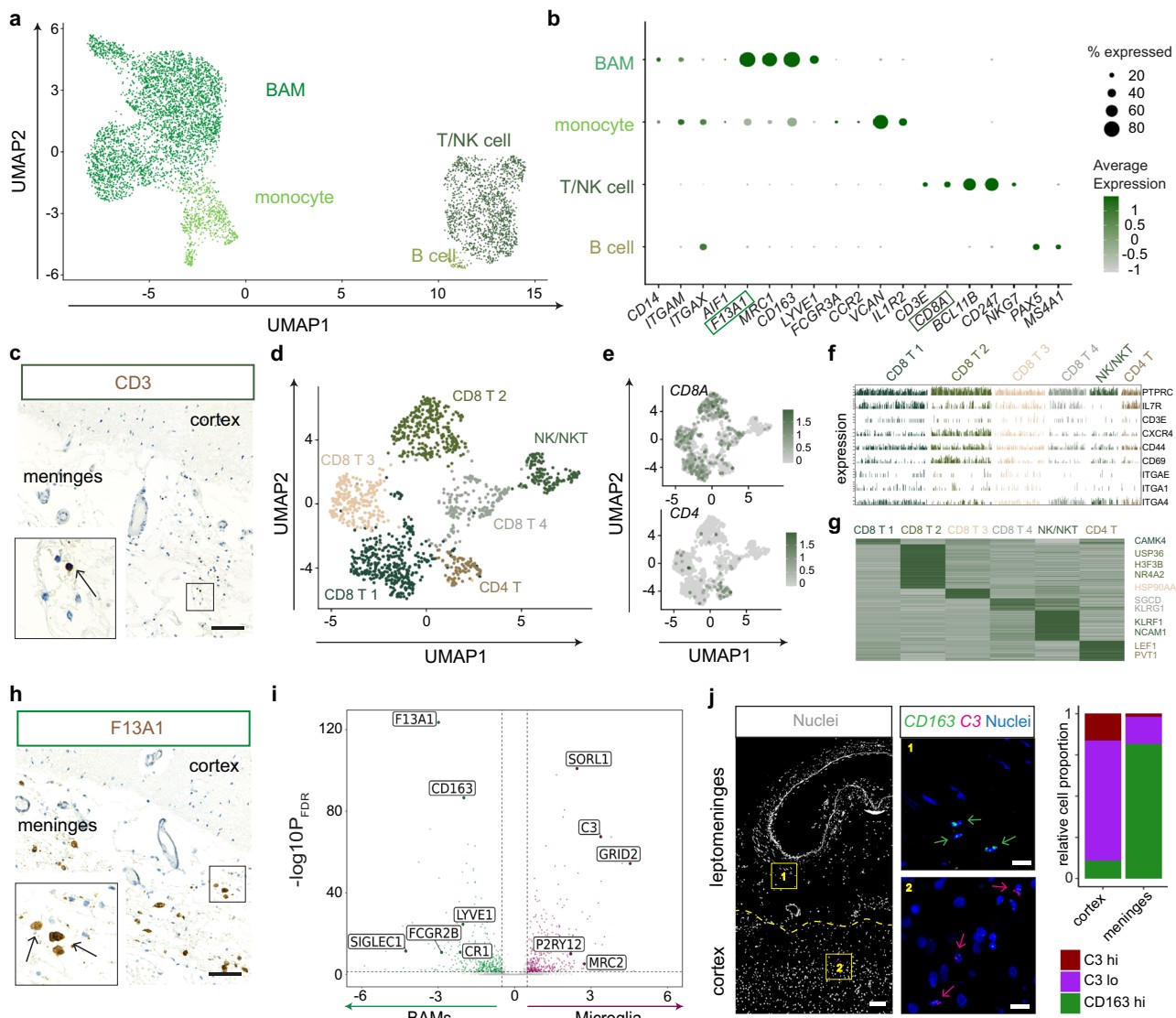

**Fig. 4 | Diverse immune cell types are detected within the human leptomeninges. a** UMAP of leptomeningeal immune cells. **b** Dot plot of canonical marker expression for the detected cell type clusters. **c** Representative chromogenic immunohistochemistry of human leptomeninges indicating protein expression of CD8. scale bar = 100 μm, n = 3 individuals (**d, e**) UMAP visualization of subclustered T cells and gene expression for *CD4* and *CD8A*. **f** Expression of T cell genes including TRM markers *CXCR4, CD44, CD69, ITGAE* per cell by T cell cluster. **g** Heatmap of the DEGs between T cell subtypes, with the representative genes annotated on the right. **h** Representative chromogenic immunohistochemistry of human

leptomeninges indicating protein expression of F13A1 scale bar = 100 μm, n = 3 individuals (**i**) Volcano plot of the differentially expressed genes distinguishing microglia from BAMs using a negative binomial generalized mixed model, Bonferroni correction; |logFC| > 0.5 and pBON <0.01 are colored. **j** A representative RNAscope image of human leptomeninges showing transcripts of BAM-enriched *CD163* and microglia-enriched *C3*. Scale bars = 100 μm (overview) 25 μm (insets). Stacked bar plots show proportions of each cell type in the parenchyma and the leptomeninges, n = 706 cells from 3 individuals.

and *ADGRG1, SLC2A5*, and *P2RY12* in microglia. In addition, we detected that *SORL1, C3*, and *GRID2* were highly expressed in microglia, whereas *F13A1* and *CD163* were in BAMs. We then performed RNAscope to validate the differential expression of *C3* and *CD163* and confirmed their highly enriched expression in microglia and BAMs, respectively (n = 706 cells from 3 individuals) (Fig. 4j). Thus, BAMs bear common yet distinct gene signatures from human microglia, indicating that the niche environment plays an important role in shaping the identity of brain resident macrophages.

## Leptomeningeal BAM subtypes and their expression of GWAS genes
Next, we sought to further interrogate the transcriptional differences among BAMs. We performed iterative clustering of the BAM subset and identified three transcriptionally distinct subtypes of BAM cells

represented in our data (Fig. 5a, b, Supplementary Data 8). We then conducted RNAscope with probes against *CD163*, expressed in all BAM clusters, and *CD83* enriched in BAM_2 cells. As expected, we detected *CD163* + *CD83*- BAM_1 and BAM_3, and *CD163* + *CD83* + BAM_2 subtypes (n = 223 cells from 3 individuals, Fig. 5c, d). To evaluate whether human BAM subtypes were transcriptionally similar to mouse BAMs, we compared the human BAM and microglial transcriptional signatures to the myeloid cell populations described in mouse border regions[12]. As expected, we found a strong correlation of gene signatures between human and mouse microglia. Notably, all three human leptomeninges BAM subtypes most resembled the mouse sub-dura/leptomeninges BAMs (Fig. 5e), suggesting the conservation of region-specific BAM gene signature across species.

To date, many AD risk loci nominated by genome-wide association studies (GWAS) reside in or near genes that are highly and/or

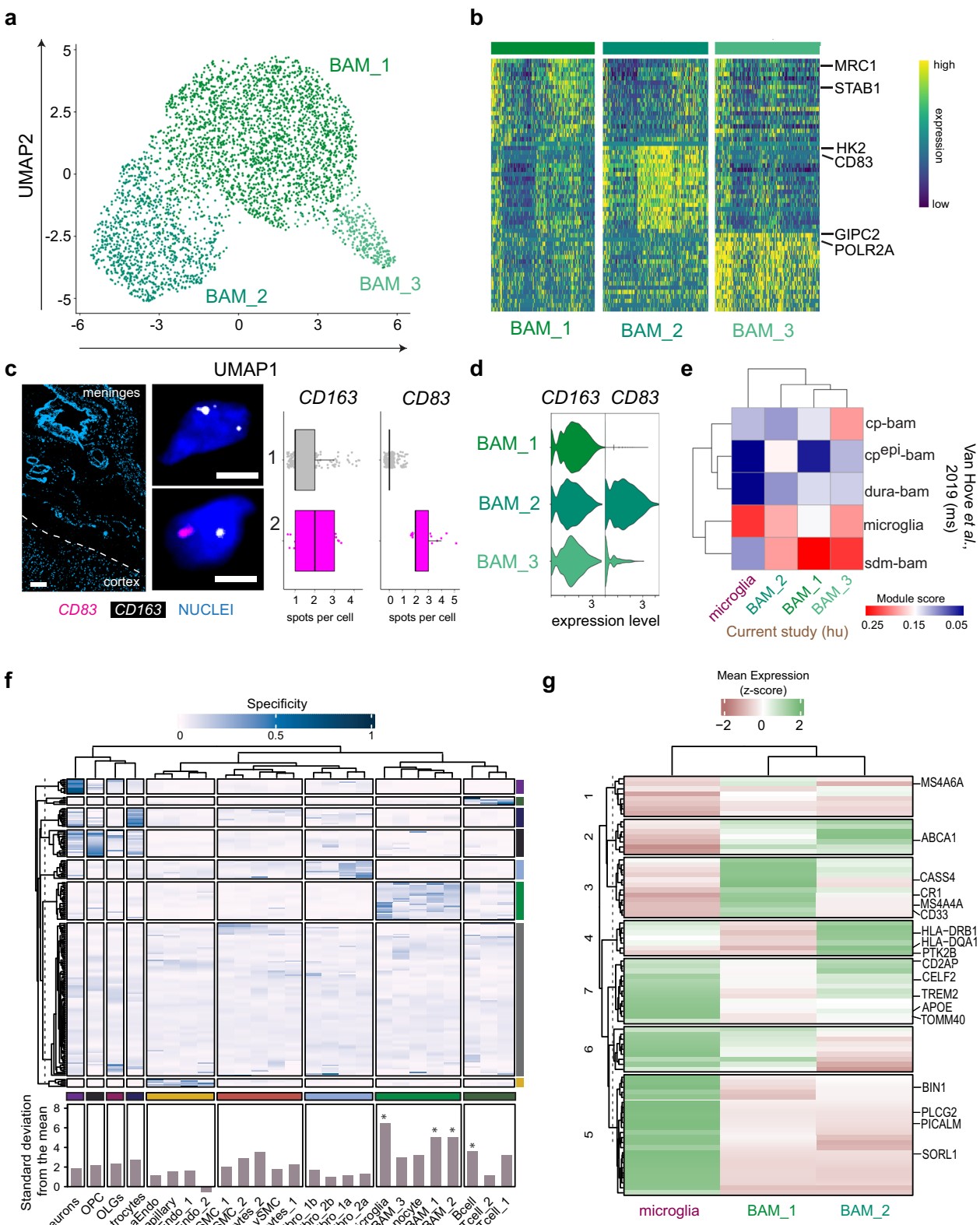

exclusively expressed in microglia[25,26]. Since our study reports the transcriptional profiles of human leptomeningeal cells, we examined the expression of GWAS genes in leptomeningeal cells and other parenchymal cell types detected in our study. We selected 212 AD GWAS genes/loci with an odds ratio greater than 1.1 based on the European Bioinformatics Institute GWAS Catalog (Supplementary Data 9). We then conducted K-means clustering that resulted in 8 distinct gene

sets. We examined the cell-type-specific expression of those genes across all cell types and detected a significant enrichment of AD GWAS genes in microglia, BAMs, and B cells (Fig. 5f). We also investigated whether GWAS genes of neurodegenerative diseases, including Parkinson's disease (PD), multiple sclerosis (MS), and frontotemporal dementia (FTD), present cell-type specific expression in leptomeningeal cell types (Supplementary Fig. 4, Supplementary Data 9).

**Fig. 5 | BAM diversity in the human leptomeninges. a** UMAP of leptomeningeal BAMs. **b** Heatmap of the expression of the top markers across each BAM subtype, with representative genes highlighted on the right. **c** Representative RNAscope image of human leptomeninges showing transcripts of *CD163* (all BAM subtypes) and *CD83* (BAM_2 subtype). Boxplots show transcript spot quantification of BAM_1 and BAM_3 (gray) and BAM_2 (pink), *n* = 706 cells from 3 individuals. All points are plotted with overlaid boxplots, center lines show the medians; box limits indicate the 25th and 75th percentiles; whiskers extend 1.5 times the interquartile range from the 25th and 75th percentiles. Scale bars = 100 μm (overview), 10 μm (insets). **d** Violin plot of *CD163* and *CD83* expression from the snRNA-seq data. **e** Heatmap of

module scores in human BAM subtypes and microglia for mouse cell-specific modules generated from Van Hove et al., 2019. cp-bam: choroid plexus bam, cpepi: epiplexus bam, sdm-bam: sub-dural meninges bam, dura-bam: dural bam. **f** Heatmap of each top-ranked AD GWAS gene's proportional expression in each detected cell type. The values on the bottom represent the relative expression of all 212 selected AD GWAS genes in each cell type. * denotes cell types with enriched expression for AD GWAS genes. **g** Heatmap of expression for the top 80 AD GWAS genes most specifically expressed in microglia, BAM_1, and BAM_2, with representative genes annotated on the right.

Notably, we found a significant enrichment of PD GWAS genes in pericytes and monocytes, MS GWAS genes in all immune cells and some endothelial cell types, whereas no cell-type enrichment of FTD GWAS genes. These results suggest that leptomeningeal cell types may be implicated in a wide spectrum of neurodegenerative diseases.

To further examine the expression of BAM and microglia-associated AD risk genes, we investigated the relative expression of 80 genes with specificity score values above 0.05 in microglia, BAM_1, or BAM_2 (Fig. 5g). We detected both common genes (clusters 2, 6, and 7) shared between cell types and unique genes primarily expressed in microglia (cluster 5), BAM_1 (cluster 3), and BAM_2 (cluster 4), respectively. At the individual gene level, *MS4A4A*, *CD33*, and *CR1* were highly expressed in the BAM_1; *HLA* genes and *PTK2B* in BAM_2; *BIN1*, *PICALM*, *PLCG2*, and *SORL1* in microglia. In addition, *APOE*, *TOMM40*, and *TREM2* were expressed in both microglia and BAM_2. In summary, like microglia, BAMs also express immune-related AD GWAS genes but with distinct expression patterns, prompting further investigations on the role of BAMs in regulating border immunity in AD pathogenesis.

### AD-associated gene modules and AD cell-type-specific DEGs in the human leptomeninges

To characterize the global transcriptional signature in AD, we performed bulk RNA-seq on postmortem leptomeninges from 44 aged individuals with varying clinical and pathological traits (AD = 23, NCI/MCI = 21; Supplementary Data 1; Supplementary Fig. 5a). We first conducted differential gene expression analysis between AD and control samples and did not detect significant differentially expressed genes (DEGs), likely due to the limited sample size. We then conducted co-expression gene network analysis using SpeakEasy[27] and identified 55 co-expressed gene modules ranging from 27 to 764 genes in size (Supplementary Data 10). To evaluate whether any modules were meninges-specific, we tested the preservation of each module in postmortem dorsolateral prefrontal cortex (DLPFC) RNA-seq data[28]. We detected 24 preserved modules and 31 non-preserved, meninges-specific modules (Supplementary Fig. 5b).

We then performed module-trait analysis. Out of 55 co-expressed gene modules, we identified 18 modules significantly associated (*p* <= 0.05) with more than one AD trait: 7 preserved modules and 11 non-preserved meninges-specific modules. (Supplementary Fig. 5c; Supplementary Data 11). Notably, 4 modules associated with more than one trait were all from meninges-specific modules. Among them, module 41 was most significantly correlated with AD clinical diagnosis. Pathway analysis showed that this module was related to ECM and complement cascades (Supplementary Fig. 5d), suggesting that matrisome and immune response in leptomeninges may be involved in AD pathophysiology.

We then assessed whether AD affects meningeal cell types transcriptionally based on the snRNA-seq data from 18 individuals (NCI/MCI = 9, AD = 9). We performed differential gene expression analysis using NEBULA, a linear mixed model approach[29]. We detected DEGs in all major cell types, especially in BAMs and fibroblasts, and most DEGs were highly restricted to one cell type, suggesting cell-type-specific transcriptional responses to AD pathophysiology (Supplementary Data 12). We detected 64 DEGs in BAMs such as *IL6R*, *TMEM39*,

*MARCH1*, and *CDK8*, which are known to be involved in inflammation, interferon signaling, immune response, and T cell activation, and cytokine release[30,31]. For fibroblast subtypes, we found that solute carrier genes, including *SLC26A2* and *SLC2A3*, were significantly reduced in Fibro_1b arachnoid barrier cells in AD (Supplementary Data 12). Because the arachnoid barrier layer is part of the blood-CSF (BCSF) barrier between the fenestrated vasculature of the dura and the CSF-filled subarachnoid space, those molecular changes may alter the BCSF barrier permeability and contribute to the immune cell infiltration and neuroinflammation in AD. Fibro_2a and Fibro_1a fibroblasts were also transcriptionally affected in AD (Supplementary Fig. 6a). Gene set enrichment analysis showed elevated ECM (collagen, laminins, and integrins) dynamics and reduced glucose metabolism in Fibro_1a and increased interferon-gamma (IFNγ) signaling in Fibro_2a (Supplementary Fig. 6b), reminiscent of the major characteristics of CNS inflammation and fibrotic scarring[32]. In summary, our snRNA-seq results suggest that immune response, ECM activation, and fibrotic scarring are important pathological processes in AD leptomeninges.

### Aβ-treated leptomeningeal cultures show the AD gene signature of ex vivo fibroblasts

As leptomeningeal cell types are exposed to CSF, we reasoned that amyloid-β in the CSF might trigger a similar transcriptional response as observed ex vivo. To test this hypothesis, we generated cell cultures from postmortem human leptomeninges (Supplementary Fig. 6c). After 2–3 passages, cells exhibited elongated bipolar or multipolar cell morphology and expressed the fibroblast-specific antigen1(FSP1), and the ECM protein collagen IV (Supplementary Fig. 6e, f). Following karyotyping, we selected 12 lines without clonal abnormalities (Supplementary Data 1, Supplementary Fig. 6d) and exposed individual cultures to amyloid-β oligomers or vehicle for 48 h, then harvested cells and performed bulk RNA-seq. We detected 20 upregulated and 2 downregulated genes (FDR < 0.05; Supplementary Fig. 6g; Supplementary Data 13). Pathway analysis of the top-ranked genes showed that ECM and immune response were significantly enriched in the amyloid-β-treated cultures (Supplementary Fig. 6h). We also detected a significant enrichment of the ex vivo fibroblast AD DEGs in the in vitro ranked gene set (Supplementary Fig. 6i). These results suggest that altered ECM and immune responses may be implicated in the amyloidogenesis of AD leptomeninges.

### Altered intercellular communications in AD leptomeninges

Previous mouse studies show that BAMs and immune cells in the meninges actively interact with non-immune cell types, forming an intricate cellular network[33]. We reasoned that the cellular communications in the leptomeningeal niche might be altered in AD due to exposure to CSF amyloid-β, tau oligomers, and other pathological stimuli. We applied CellChat algorithm[34] to infer intercellular communications among leptomeningeal cell types and between parenchymal and leptomeningeal cell types in our dataset.

To infer intercellular communications among ten leptomeningeal cell types, we first examined differentially expressed ligands and receptors for all cell groups, resulting in 27 significant ligand-receptor pairs. We then quantified intercellular communications and detected a

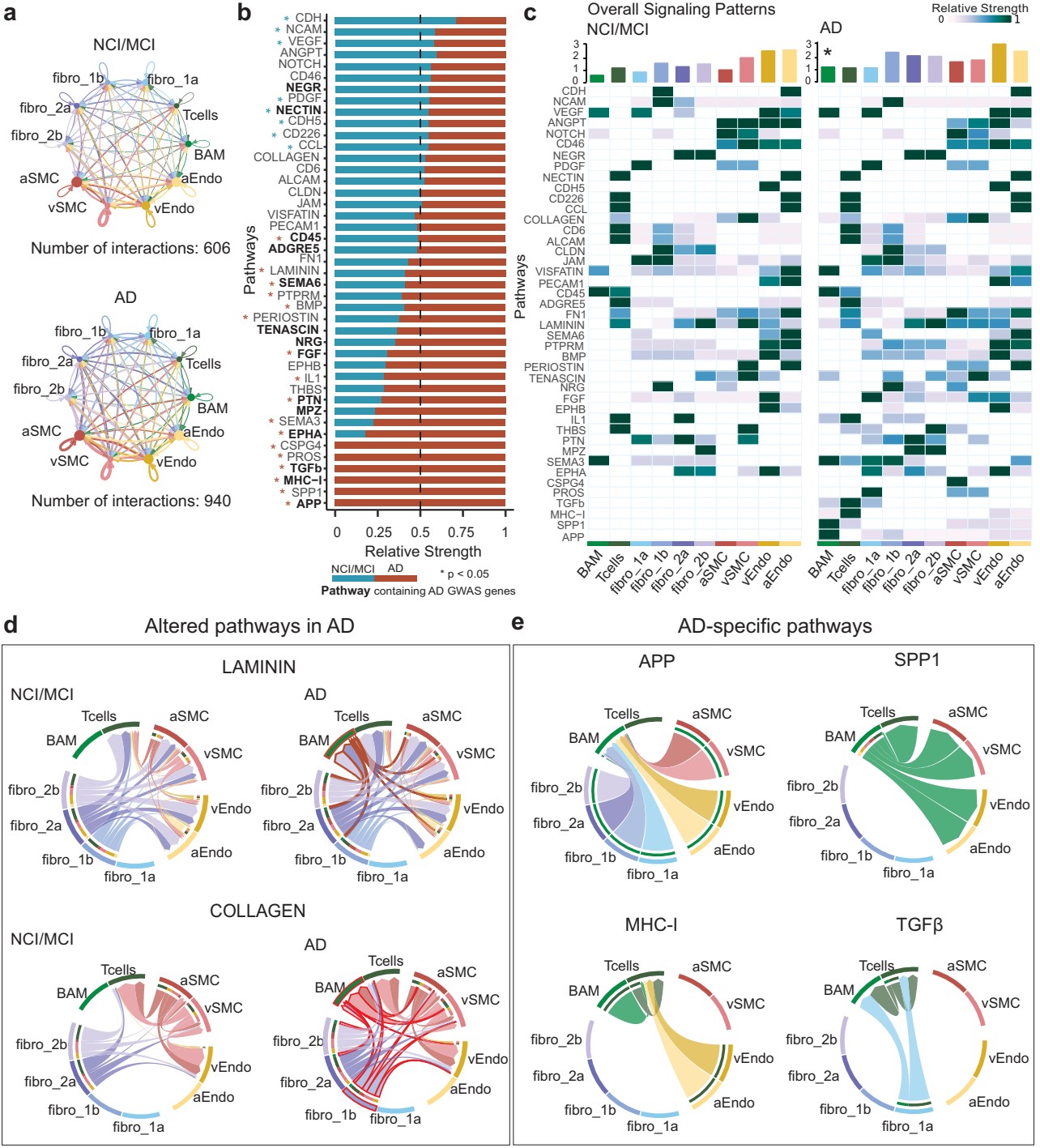

**Fig. 6 | Altered intercellular communications in AD leptomeninges. a** Circular plots of all inferred intercellular interactions in the control (NCI/MCI) and AD groups. Autocrine (loops on top of each cell group) and paracrine (connections between cell groups) communications are displayed in each circular plot. **b** The relative strength of 37 active signaling pathways in the NCI/MCI and AD group. Pathways annotated with blue and red stars are downregulated and upregulated in AD, respectively. Pathways containing AD GWAS genes are bolded. Two-sided

Wilcoxon Rank-Sum $p < 0.05$. **c** Heatmap showing cell-type-specific signaling patterns in NCI/MCI and AD. The bar plots on the top show the overall signaling activity of each cell type.* denotes significantly increased activity. **d, e** Circular plots show the altered laminin and collagen ECM pathways in AD (**d**) and AD-exclusive pathways such as APP, SPP1, MHC-I, and TGFβ (**e**). Each arrowed line represents the signaling flow from the sender to the receiver cell subtype. Aberrant interactions are highlighted in red for AD.

greater number and strength of communications in AD (Fig. 6a, Supplementary Fig. 7a, b). To further evaluate signaling activities, we performed the differential analysis of intercellular communications and detected greater autocrine and paracrine activities in/between most cell types in AD (Supplementary Fig. 7b). After annotating those intercellular communications, we categorized the ligand-receptor pairs into 43 active signaling pathways, of which 6 signaling

pathways were exclusively active in AD (Fig. 6b). Of 37 shared signaling pathways, 20 were differentially expressed in AD. Next, we constructed the signaling communication patterns of all cell types and calculated the overall communication strength per cell type. As expected, we detected a trend for increased communications in most AD cell types, especially BAMs (Fig. 6c). We then divided the signaling pathways into incoming and outgoing signaling and calculated the strength by cell

type accordingly. Notably, we observed greater outgoing Fibro_1b signaling and incoming BAM signaling in AD, respectively (Supplementary Fig. 7c, d). When we examined a couple of known ECM pathways, we observed aberrant communication patterns of the laminin, collagen, and tenascin pathways in AD, centering around fibroblasts and BAMs (Fig. 6d, Supplementary Fig. 7e, f). In addition, known immune-related pathways, such as APP, SPP1, MHC1, and TGFβ signaling, were found exclusively active in AD (Fig. 6e).

Next, we leveraged the parenchymal cells captured in our study and assessed the intercellular communications between leptomeningeal and parenchymal cells. For this analysis, we collapsed cell types into four major cell types: parenchymal cells, leptomeningeal vessel cells, leptomeningeal immune cells, and leptomeningeal fibroblasts. We consistently detected stronger interactions in AD at the parenchyma-meningeal interface (Supplementary Fig. 8a). Interestingly, we observed elevated reciprocal interactions between parenchymal and leptomeningeal immune cells in AD (Supplementary Fig. 8b, c). Among the 25 active pathways, 8 were significantly lower, and 11 were higher or exclusively active in AD (Supplementary Fig. 8d). Overall, leptomeningeal immune cells in AD demonstrated significantly more interactions with other cell types, including parenchymal cells (Supplementary Fig. 8e). Taken together, our CellChat results show increased overall intercellular activities and altered communication patterns in AD leptomeninges, suggesting overactive ECM and immune networks an important part of AD pathophysiology.

## Discussion

We present the single-nuclei characterization of 42,557 cells from isolated postmortem human leptomeninges. Our study provides a comprehensive transcriptomic atlas of the aged human leptomeninges, revealing rich cell type diversity within the stromal and immune cell types.

For vascular cells, we discover unique vEndo and vSMC clusters in the leptomeninges, in line with the previous report that the larger pial veins have specific circumferentially oriented smooth muscles that control blood flow and drain the cerebral cortex[35]. By integrating with snRNA-seq data from parenchymal vessels[14], we construct the transcriptional zonation, recapitulating a complete continuum of anatomical arteriovenous axis from leptomeningeal arteries to parenchymal arterioles, capillaries, venules, and back to leptomeningeal veins.

For fibroblasts, we detected three subtypes (Fibro_1a, Fibro_2a, Fibro_2b) that are transcriptionally similar to those found in the parenchymal studies[14,16] but with different representations of each cell type. The transcriptional similarity is not entirely surprising because of three major anatomical features of parenchyma and leptomeninges: the continuation of perivascular fibroblasts around small and large vessels, the covering of the pia mater around arterioles/arteries in the parenchymal perivascular space and the extension of the pia mater into the brain sulcus. Importantly, we discover a leptomeninges-specific fibroblast subtype, the Fibro_1b arachnoid barrier cells. Because tight junctions between the arachnoid barrier cells create a BCSF barrier that exhibits selective impermeability to immune cells and macromolecules, the changes in solute carrier genes, including *SLC26A2* and *SLC2A3*, identified in our study indicate altered BCSF barrier function in AD.

For immune cells, we detect diverse T cell subtypes in the leptomeninges, primarily CD8+ cells, expressing TRM core genes. Because the leptomeninges are exposed to CSF-borne antigens, T cells may interact with antigen-presenting cells to provide site-specific immunity. Concordantly, CD8+ TRMs have been found to be populated in the parenchyma and perivascular space of human brains[23]. Because the subarachnoid space (SAS) connects with the perivascular space, CD8+ TRMs in those compartments may work synergistically to conduct border immunity. The origin of T cells and monocytes in our study

remains to be investigated. Since the vasculature in the pia is not fenestrated, the immune cells could either breach the tight junctions of the arachnoid mater from the dura or cross the walls of the leptomeningeal post-capillary venules to reach the SAS. Such T cell trafficking and infiltration routes have been reported in animal models of neuroinflammation, such as multiple sclerosis[36,37]. Because the leptomeninges tissue in our study was from aged individuals, it is possible that the presence of T cells is due to a general weakening of brain barrier functions with aging. Indeed, T cells have been recently shown to infiltrate the subventricular zone of old mice and humans[38]. It is worth noting that CSF from healthy individuals harbors tissue memory CD4+ and CD8+ T cells[37,39–41] and a small fraction of those T cells express genes related to tissue residence[40,42], indicating a release of TRMs from surrounding tissue, such as leptomeninges, to the CSF. Interestingly, a recent elegant study shows effector T cells constantly traffic between the leptomeninges and the CSF, where activated T cells attached to the leptomeninges and non-activated cells are released into CSF[37]. Therefore, T cells detected in our study may receive activation and adhesion signaling, thereby retaining in the leptomeningeal niche.

Our study presents the transcriptional profiles of BAM subtypes in the aged human leptomeninges. Notably, BAMs display 780 differentially expressed genes from parenchymal microglia, indicating that the local niche signals may shape the gene signatures of BAM versus microglia and instruct their region-specific identities. In addition, BAMs and microglia both express more AD GWAS genes than other cell types, highlighting the important role of the immune component in AD development and the need to investigate BAMs, this understudied immune population. Interestingly, we also found that many AD GWAS genes are differently expressed among BAM subtypes and microglia, strongly suggesting their overlapping and distinct functions in contributing to the AD process. Because leptomeningeal BAMs have been shown to be repopulated by blood monocytes or through their self-renewal capacity after deletion[36,43], it is also possible that the monocytes detected in our study replenish the BAMs through a peripheral blood mechanism.

The essential roles of meningeal immunity have been linked to traumatic brain injury, stroke, infection, multiple sclerosis, aging, and AD[5,44,45]. We report widespread cell type-specific differential gene expression in the leptomeninges of AD individuals, especially in BAMs and fibroblasts. Those dysregulated genes in BAMs strongly suggest an aberrant immune response in AD leptomeninges. Furthermore, the DEGs in fibroblasts are related to ECM dynamics, immune response and cytokine signaling. Our CellChat analysis infers altered intercellular ECM and immune pathways in AD, converging on BAMs and fibroblasts. Therefore, it is possible that the immune response in leptomeninges triggers fibrotic and ECM response in AD leptomeninges, as reported in the mouse model of multiple sclerosis[32]. In addition, our meningeal cultures demonstrate an enriched gene signature of ex vivo AD fibroblasts upon amyloid-β treatment, indicating that exposure to amyloid-β may directly induce the ECM and immune responses.

Our study catalogs the major immune and stromal cells in the human aging leptomeninges, revealing their transcriptional profiles and elucidating the molecular difference among microglia and BAM cells. The enriched AD GWAS gene expression in BAMs underlies the important roles of those cells in border immunity and defense mechanisms. Our study also uncovers a potentially novel mechanism, leptomeningeal immunity, in AD pathophysiology. Supporting evidence from mouse studies already suggests that BAMs play a role in vascular Aβ clearance[25,46–48]. Therefore, targeting the immune and stromal cells in the meninges may provide a promising approach to modulating border immunity in fighting AD and other neurological diseases.

## Limitations of study

There are several limitations of this study. First, our samples are from aged individuals, so the gene profiles of leptomeningeal cell types may reflect some aging processes. Future samples from a wide range of age groups will help to elucidate the aging effect on border cell types. Second, although we have a standard protocol to dissect the leptomeninges, some technical variations could remain. Those variations, along with our limited sample size, may lead to over- or under-sampling of specific cell types, especially those rare ones. Increased sample size and rigorous QC will help us to resolve more rare cell types in the future. Third, we do not uncover spatial distribution patterns of most cell types except for Fibro_1b arachnoid barrier cells due to the limited workable antibodies and probes. Future studies using the spatial transcriptomics approach will help to address this issue and provide the spatial context of specific cell types and their relationships to anatomical structures and AD pathologies.

## Methods

### Human postmortem leptomeninges tissue collection

Human leptomeninges tissue was collected from 46 individuals from the Religious Orders Study or Rush Memory and Aging Project (ROS-MAP) study[49]. Both studies were approved by an Institutional Review Board (IRB) of Rush University Medical Center and in accordance with the criteria set by the Declaration of Helsinki. All participants signed an informed consent and Anatomic Gift Act. All autopsies in this study were performed at the Rush Alzheimer's Disease Center following a PMI of ~8.5 h. All autopsy procedures and tissue collection were performed by staff blinded to pathological or clinical diagnoses. We adopted a published protocol for leptomeninges dissection and cryopreservation[46]. Briefly, the skull with the adhered dura was first removed, then the brainstem was severed at the level of the mammillary bodies. The cerebrum with attached leptomeninges was hemisected to ensure that the leptomeninges were intact, as assessed by the presence of blood vessels and arachnoid trabeculae. The leptomeninges were then stripped from the motor cortex region using forceps, and a $1 \times 6$ cm piece was immediately placed in ice-cold collection media (2% FBS, 1% Sodium pyruvate, 1% Sodium pyruvate in PBS) and transferred to the lab on ice for further processing. After brief washing with PBS, leptomeninges were stored in a cryoprotectant solution (11% sucrose and 10% DMSO in sterile water) in LN2 for future cell line derivations and RNA-seq experiments. 12 leptomeningeal cell lines were used for our culture experiments. 44 leptomeninges samples were used for RNA-seq, and a subset of 18 samples were used for snRNA-seq. The single nuclei samples were selected based on clinical diagnosis of Alzheimer's dementia ($n = 9$) and individuals with no cognitive impairment (NCI) or mild cognitive impairment (MCI), served as the control group ($n = 9$); the two groups were balanced for age, sex, and cerebral amyloid angiopathy load. Diagnostic procedures have been previously reported[50–52]. Analyses also used pathologic AD and cerebral amyloid angiopathy (CAA) assessed as previously described[53,54]. Detailed phenotypic information can be found in Supplementary Data 1.

### Sample preparation, library construction, sequencing, and data processing for total RNA-seq

Frozen leptomeninges tissue and leptomeningeal cell lines were homogenized in DNA/RNA shield buffer (Zymo, R1100) with 3 mm beads using a bead homogenizer. RNA was subsequently extracted using a Chemagic RNA tissue kit (Perkin Elmer, CMG-1212) on a Chemagic 360 instrument. RNA was concentrated (Zymo, R1080), and RNA quality number (RQN) values were calculated with a Fragment Analyzer total RNA assay (Agilent, DNF-471). RNA concentration was determined using Qubit broad-range RNA assay (Invitrogen, Q10211) according to the manufacturer's instructions. 500 ng total RNA was used as input for sequencing library generation, and rRNA was

depleted with RiboGold (Illumina, 20020599). A Zephyr G3 NGS workstation (Perkin Elmer) was utilized to generate TruSeq stranded sequencing libraries (Illumina, 20020599) with custom unique dual indexes (IDT). Library size and concentrations were determined using an NGS fragment assay (Agilent, DNF-473) and Qubit ds DNA assay (Invitrogen, Q10211), respectively, according to the manufacturer's instructions. Libraries were normalized for molarity and sequenced on a NovaSeq 6000 (Illumina) at 40–50 M reads, 2x150bp paired-end.

RNA-seq reads were aligned to the GRCh38 (hg19) reference genome via STAR 2.4.2a with comprehensive gene annotations from Gencode v27. QC metrics were calculated from PicardTools v1.128, and STAR. In the quantification pipeline, transcript raw counts were calculated by Kallisto (v0.46). Transcript counts were aggregated at the gene level to obtain gene counts separately in mRNAs and pre-mRNAs. Samples were excluded if the total reads mapped were less than 5 million. Data were further processed in R (v4.0.5).

We filtered out low expressed genes (counts <1000), normalized for library size by using trimmed mean of M-values (TMM), and transformed the data to log2-CPM (counts per million) using voom function from limma package (v3.46). Confounding factors, including biological, technical, and Picard-reported sequencing metrics (age, sex, RIN, RNA library batch, percentage of intergenic bases, percentage of intronic bases, mean insert size), were regressed out.

### Bulk RNA-seq analysis and module generation

For the detection of differentially expressed genes, we fitted a linear regression model with binary clinical or pathological diagnosis as predictors using the limma package (v3.46). The significance level was set at a false discovery rate (FDR) ≤ 0.05. We applied the SpeakEasy[27] consensus-clustering algorithm to the data to identify modules. To annotate the modules, we performed enrichment analysis for Reactome terms with functions from R package clusterProfiler[55] (v4.1.4). WGCNA package[56] (v1.70-3) was used to test the preservation of these meninges modules in dorsolateral prefrontal cortex (DLPFC) data from Mostafavi et al., 2019[28]. Module preservation was assessed after 1000 permutations, and the threshold for preserved modules was set at $z_{summary} > 7.5$ based on package creators' recommendations. One-way analysis of variance (ANOVA) was used to identify which modules were associated with AD-related traits, i.e., clinical diagnosis, pathological diagnosis, cerebral amyloid angiopathy (CAA) pathology, amyloid load, neurofibrillary tangles, Braak stage, cognitive decline. Significance was set at $p_{FDR} \leq 0.05$.

### Isolation of nuclei from postmortem leptomeninges

We employed a modified VINE-seq[14] protocol to facilitate vascular nuclei extraction. Briefly, leptomeninges tissue was thawed in homogenization buffer [250 mM sucrose, 25 mM KCl, 5 mM MgCl2, 10 mM Tris pH8, 1 μM DTT, 15 μM actinomycin, 0.2U/μl RNAse inhibitor], then pressed through a 100 μm cell strainer using the back end of a 5 ml syringe plunger, wetting the filter intermittently. The cell suspension was centrifuged at $300 \times g$ for 5 min at 4 °C. The pellet was resuspended in fresh homogenization buffer, transferred to a Dounce homogenizer, and homogenized 10 times slowly with pestle B. Nuclei were filtered through 40 μm flowmi strainers and fixed in up to 80% ethanol for 30 min at 4 °C. Fixed nuclei were washed in an equal volume of wash buffer [PBS, 2% BSA, 0.2U/μl RNAse inhibitor] and centrifuged at $800 \times g$ for 5 min at 4 °C. Fixed nuclei were washed three additional times and filtered through 40 μm flowmi strainers. Nuclei were counted using trypan blue on a Countess (Invitrogen).

### Single nuclei RNA sequencing and data processing

We prepared libraries using 10x Genomics 3' single-cell gene expression assays v3 and sequenced them on a NovaSeq 6000 to a depth of 50k reads/cell. Following sequencing and FASTQ generation, raw count matrices were produced using CellRanger v6.0.1. RNAs were

mapped to the transcriptome, assigning transcripts to individual cells and removing duplicate reads. The resulting count matrices were processed using the Seurat[57] v4.2.0 package in R. Cells containing <500 or > 5000 genes and/or > 5% mitochondrial RNA reads were removed. The Seurat SCTransform[58] function was used to normalize and scale the UMI counts for each individual library based on regularized negative binomial regression and regress out mitochondrial read percentage. Principal component analysis (PCA) was performed on the union of the top 3000 variable genes by sample. Integration of samples across donors was performed using Harmony[15] (v0.1.0). We used Seurat to perform UMAP dimension reduction on the top 20 Harmony embeddings and used the findNeighbors and findClusters functions to identify clusters based on a shared nearest neighbor (SNN) clustering algorithm. The data was manually inspected for cell doublets and those containing mixed cluster signatures were removed. For subsequent interrogation within each cell type, individual cell clusters were subsetted and re-clustered using the functions mentioned above. We excluded the cells from one immune cluster that expressed markers of neutrophils as it was exclusively present in a single sample and therefore could not be replicated across participants.

### Marker gene identification and cell type annotation

To identify gene markers for each cluster we performed a Wilcox rank sum-test using Seurat's FindMarkers, testing each population against all other populations. Clusters were then annotated using canonical markers. The endothelial clusters were initially identified by co-expression of *PECAM1*, *CLND5*, *FLT1*; mural cells by co-expression of *ACTA2*, *MYH11*, *CSPG4*, and *COL4A1*; fibroblasts by co-expression of *DCN*, with either *SLC4A4* or *LAMA2*; and immune cells by co-expression of *PTPRC* with either *MRC1* and *F13A1* or *PARP*8 and *THEMIS*. All clusters showed an absence or significantly lower expression of the key markers of other cell types. Cell-(sub)type proportional differences between AD and the NCI/MCI group were interrogated using the propeller function from R package speckle (v0.0.3). Cell subtypes with unique transcriptional signatures within broader clusters were numbered, Fibro_1a, Fibro_1b, etc. and markers are reported in Supplementary Data 2, 5, 6 and 8. For endothelial and mural cells, the integrated datasets were annotated using cell type definitions from Yang et al.[14]. To identify differentially expressed genes between cell subtypes, we used the R package nebula[29] (v1.2.1). We fitted a negative binomial mixed model correcting for age and sex and adjusting the random effect of the donor and the cell UMI number. We filtered out genes that were expressed in less than 10% of the cells in each subtype. The threshold for differential expression was set at a natural log fold change (absolute value) ≥ 0.5, and Bonferroni corrected *P* value ≤ 0.01. R package EnhancedVolcano[59] (v1.14.0) was used for visualization of the differentially expressed genes.

### Single nuclei dataset integration for endothelial, mural and fibroblast cells

Data from Yang et al.[14] was kindly provided as a Seurat object by the authors. Normalization and scaling of the raw data, as well as detection of variable features, were performed on the sample level with the SCTransform function from Seurat. The individual sample objects were then merged into a single object with similarly preprocessed leptomeningeal data. For integration, equivalent cell types from both studies were subsetted, and PCA was performed on cell type-specific datasets using merged sample-level variable features. Datasets were integrated using Harmony[15] (v0.1.0), accounting for donors and regions. Further parameters in RunHarmony were set as follows: lambda = c (0.1,0.1), tau = 700, theta = c (2,2). Cells were re-clustered using the first 10 dimensions of the Harmony embedding to identify integrated subclusters and visualized using UMAP.

### Mouse and Human vSMC comparison

The murine vSMC gene expression module was generated based on the human orthologs of the 41 mouse vSMC genes listed in Fig. 5B of Muhl et al.[20]. The module score was calculated for each leptomeningeal SMC cell using the AddModuleScore function in Seurat and then plotted on the joint UMAP.

### Trajectory analysis of endothelial and mural cells

Cell trajectory order of the integrated mural and endothelial cell datasets was conducted using pseudotime analysis in Monocle3[60]. Endothelial cells formed one trajectory, whereas mural cells formed two trajectories with vSMCs separated from the aSMC-pericyte continuum. Due to the possibility of batch effects originating from the different studies being conflated with cell types, we used only the leptomeningeal dataset to perform spatial auto-correlation analysis identifying trajectory-variable genes[61]. We fit splines to smooth gene expression along the trajectory for genes with Moran's I > 0.1, clustered genes by expression pattern and visualized these using heatmap.

### Gene ontology network analysis for fibroblast subtypes

The Gene ontology network for the fibroblast subtypes was constructed by using ClueGO[62] (v2.5.9) plug-in of Cytoscape (v3.9.1). Top marker genes from each fibroblast subtype were used as the input. We used GO_BiologicalProcess-EBI-UniProt-GOA-ACAP-ARAP_22.05.2022 ontology. A right-sided hypergeometric test was used to determine ontology enrichment with Benjamini-Hochberg multiple testing correction (PBH ≤ 0.01). Parameters for network construction were set as follows; GO level range: 3-8, Kappa score: 0.4, the minimum number of genes included in term = 3, the minimum percentage of genes included in term = 4%.

### Mouse and Human BAM comparison

Raw single-cell data from van Hove et al.[10] was downloaded from GSE128855, and processed as described by the authors. Mouse bam populations from different brain border regions were identified, and gene markers between bam subtypes were determined using ROC analysis implemented in Seurat's FindMarkers. To compare mouse and human cells, human homologs of bam subtype genes with roc > 0.7 were used to generate gene expression modules for each mouse subtype and calculated per leptomeningeal cell using the AddModuleScore function in Seurat.

### Cell-type-specific differentiation gene expression in AD

To identify which genes were differentially expressed in each cell type between the AD and the control group, we fit a linear mixed model that predicted the expression level of each gene per cell based on AD diagnosis, corrected for age, sex, and mitochondrial gene ratio. R package nebula[29] (v1.2.1) was used to implement the model, including parameters for a zero-inflated negative binomial distribution, the random effect of the donor, and the cell UMI number. We filtered out genes that were expressed in less than 5% of the cells in each subtype. The threshold for differential expression was set at a natural log fold change (absolute value) ≥ 0.3, and Bonferroni corrected *P* value ≤ 0.01. We opted for a stringent threshold as the nebula is reported to produce type II error/inflated *p* values when applied in smaller sample sizes. Visualizations of the number of differentially expressed genes were generated with the R package ComplexUpset (v1.3.3). Gene set enrichment analysis against REACTOME pathways was performed with R package fgsea[62] (v1.22.0). Significance levels were set at a false discovery rate (FDR) ≤ 0.05.

### Leptomeningeal cell line derivation

We adopted a previously reported protocol to generate leptomeningeal cell lines[63]. Briefly, cryopreserved leptomeninges tissue was thawed, dissected into 2–4 mm² pieces with larger vessels removed,

and then digested in 1000 U/mL collagenase type IA solution (Gibco) for 90–120 min at 37 °C with agitation every 30 min. Leptomeningeal cells were then washed, filtered, and plated into a 6-well plate coated in 0.1% gelatin (Sigma Aldrich). Once plated, cell lines were maintained in complete culture media (Dulbecco's Modified Eagle's Medium, 10% fetal bovine serum, 1% Amphotericin B and 1% penicillin/streptomycin solution) under standard cell culture conditions (37 °C, 5% $CO_2$ and 95% humidity). Karyotyping was performed by Cell Line Genetics LLC (Madison, WI). Only cell lines without any clonal aberrations were used for downstream in vitro experiments.

### H&E staining, immunohistochemistry, and immunofluorescent staining

FFPE meningeal sections were pre-heated for 30 min at 60 °C in a dry oven and then rehydrated using a series of xylene and ethanol dilutions before rehydrating in DI water. For H&E staining, sections were submerged in hematoxylin solution for 13 min at room temperature and then rinsed in tap water for 5 min. Sections were then submerged in 0.2% ammonia water solution for 5 s, rinsed in tap water for 5 min, then submerged in eosin solution for 25 min at room temperature. Following staining, sections were dehydrated prior to mounting. Imaging was performed using a ×20 objective on Eclipse Ti2-E microscope (Nikon). For HRP-DAB chromogenic immunohistochemistry, meningeal sections were completed with the Bond-Rx Fully Automated Research Stainer (Leica Biosystems) following the manufacturer's User Manual for BDZ 11. Autofluorescence was quenched with a 30-second exposure to True-Black Lipofuscin Autofluorescence Quencher (23007, Biotium) and washed with PBS before mounting. Imaging was performed on either a CellInsight CX7 LZR High Content Analysis Platform (ThermoFisher Scientific) or an Eclipse Ti2-E microscope (Nikon) at 20x and 40x. Antibodies are listed in Supplementary Data 14.

For immunofluorescence staining of cultures, meningeal cells were fixed with 4% paraformaldehyde in PBS at room temperature (RT) for 15 min and blocked in 5% donkey serum in PBS at RT for 30 min. Primary antibodies were used at 1:300 for both anti-laminin (Millipore Sigma; AB19012) and anti-collagen (Millipore Sigma; AB769). Secondary antibodies were used at 1:500 for Alexa Fluor 488 and Alexa fluor 594 (Invitrogen). Hoechst (Invitrogen) was used for nuclei visualization. Images were acquired with CellInsight CX7 LZR (Thermo scientific).

### In situ hybridization

In situ hybridization of human FFPE meningeal sections was performed with a Bond-Rx Fully Automated Research Stainer (Leica Biosystems) following the RNAScope LS Multiplex Fluorescent Reagent Kit (322800, ACD) User Manual for BDZ 11 with the following modifications. Sections were baked for 30 min at 60 °C prior to dewaxing for 30 s at 72 °C. Sections were antigen retrieved using HIER with a pH 6.0 citrate buffer solution (AR9961, Leica Biosystems) for 20 min at 100 °C, washed, and treated with ACD Protease III for 15 min at 40 °C prior to probe hybridization for 2 h at 40 °C. TSA Plus fluorescein, Cy-3, and Cy-5 (Akoya Biosciences) were diluted 1:500 for signal development. Autofluorescence due to age-related endogenous lipofuscin was quenched with a 30-second exposure to True-Black Lipofuscin Autofluorescence Quencher (Biotium), and slides were washed with PBS before mounting. Imaging was performed on either a CellInsight CX7 LZR High Content Analysis Platform (ThermoFisher Scientific) or an Eclipse Ti2-E microscope (Nikon) at ×20 and ×40.

### QuPath Semi-quantitative analysis

RNAScope and IF staining were imaged at ×40 magnification for 10 × 10 fields using CellInsight CX7 LZR High Content Analysis Platform (ThermoFisher Scientific) with subsequent stitching and merging into composite TIF images. RNAScope probes are listed in Supplementary Data 14. Composite TIF images were uploaded to QuPath (v0.3.2) for semi-quantitative analysis. Regions of interest were identified and

selected in the composite TIF images, and spot detection was determined by thresholding parameters, including background estimation, cell and spot size, and intensity. Cells were defined using QuPath Cell Detection tool with various nucleus parameters, including 0.5 μm requested pixel size, 0μm background radius, 0μm median filter radius, 1μm Sigma value for Gaussian filter, 5μm² minimum nucleus area, 400 μm² maximum nucleus area, and 1700 intensity threshold. Cell parameters include a cell expansion of 2 μm from the cell nucleus and general parameters include smoothing of the detected nucleus/cell boundaries. For fibroblasts, spot-level data was filtered to those containing a minimum of 2 spots/cell with at least 1 spot from SLC7A2 or LAMA2 channels (Fig. 3). For BAMs/microglia spot-level data was filtered to those containing a minimum of 2 spots/cell (Fig. 4). For BAM subtypes spot level data was filtered to those containing a minimum of 2 spots/cell with at least 1 spot from CD163 (Fig. 5). Cells containing more than 8 spots for any single channel were removed. kmeans clustering was then performed to group RNAscope signal patterns across cells. These RNAscope-based clusters were manually compared to cell-type clusters from single nuclei sequencing data.

### Flow cytometry

Cell cultures were washed with PBS and singularized using trypLE. Cells were resuspended in a flow buffer (PBS + 0.5% BSA) and counted. Cells were blocked and incubated with FSP1 antibody on ice for 30 min. Dead cells were excluded with 7AAD. Samples were analyzed using a Sony SH800 flow cytometer, and data was processed using FlowJo v10.

### Amyloid-β oligomer treatment

Hexafluoroisopropanol (HFIP)-treated Aβ1-42 peptide (Echelon Biosciences) stocks were prepared following Stine's protocol[64]. Treated Aβ1-42 peptides were dissolved in dimethyl sulfoxide (DMSO; Sigma) and then sonicated for 10 min. Prepared Aβ1-42 aliquots were further diluted in cold Dulbecco's Modified Eagle Medium/Nutrient Mixture F-12 (DMEM/F12; Thermofisher) media to a final concentration of 100 μM and were stored at 4 °C for 24 h. Leptomeningeal cell lines were exposed to either 10 μM oligomeric Aβ 1-42 or vehicle for 48 h, then washed with PBS and resuspended in DNA/RNA shield buffer (Zymo) for storage prior to RNA sequencing.

### Meningeal cell line RNA sequencing

Total RNA sequencing libraries were generated and preprocessed for bulk meninges tissue. We fitted a linear regression model to identify differentially expressed genes upon Aβ treatment and visualized the results with functions from the R package EnhancedVolcano[59] (v1.14.0). Gene set enrichment analysis on the ranked gene set was performed against the REACTOME pathways with functions from fgsea[65] package (v1.16). The enrichment of the AD-associated up- and downregulated genes from ex vivo fibroblasts, endothelial, mural and immune cells in the ranked in vitro gene set was performed with the GSEA function from clusterProfiler[55] (v4.1.4). The significance level for all analyses was set at a false discovery rate (FDR) < 0.05.

### GWAS gene expression in different cell types

We collected a list of AD GWAS genes ($n = 829$) from the European Bioinformatics Institute GWAS Catalog[66] and selected gene loci with an odds ratio (OR) higher than 1.1 for downstream analysis ($n = 212$) (Supplementary Data 9). Cell subtype enrichment of the GWAS gene sets was performed with expression-weighted cell type enrichment analysis (EWCE; v1.4.0)[67]. Normalized single nuclei counts were used as input for the analysis. A bootstrap test with 10,000 permutations and controlled gene size was applied to test for cell type enrichment using all expressed genes as background. Of the 212 originally input GWAS genes, 186 passed filtering and were further considered for the bootstrap test. The standard deviation from the mean of the bootstrapped values for each cell type was used as the outcome, and significance was

defined after Bonferroni correction ($p \leq 0.05$). Additionally, proportional cell subtype-specific gene expression was calculated, with the sum of a gene's expression across all cell subtypes being 1. Based on kmeans clustering, 8 clusters of genes were identified and were annotated to the strongest/more specific cell subtype (Supplementary Data 9). Gene ontology of gene clusters was performed using Metascape[68]. For closer interrogation of the GWAS genes expression within the myeloid cluster, we subsetted 80 genes (Supplementary Data 9) specifically expressed in microglia, BAM_1 or BAM_2 by filtering for specificity values above 0.05, which reflected the fourth quartile of specificity value in each cell type.

In addition, we complied gene sets from the most comprehensive GWASs of Parkinson's disease[69] ($n = 66$), Multiple Sclerosis[70] ($n = 255$), and Frontotemporal Dementia[71] ($n = 34$) and performed the same analysis.

## CellChat analysis
Intercellular communication patterns among different cell types were inferred, visualized, and analyzed from the generated snRNA-seq data using CellChat[34] (v1.4.0). We subsetted the normalized counts into control and AD and processed them in parallel as a CellChat object, following the official workflow. For each of the datasets, we used the preprocessing functions 'identifyOverExpressedGenes', 'identifyOverExpressedInteractions', and 'projectData' with standard parameters set. We utilized all available ligand-receptor interactions from CellChatDB. For the Inference of cell-cell communication network in control and AD, we applied the functions 'computeCommunProb', 'computeCommunProbPathway', and 'aggregateNet'. Finally, the function 'netAnalysis_computeCentrality' was applied to identify dominant senders and receivers in the intercellular communication networks created. At this point, we combined the NCI and AD CellChat objects for downstream comparison analysis. Differential number and weight of interactions were calculated. Differential signaling pathway activities were identified with a paired Wilcoxon test by comparing the sum of communication probability among all pairs of cell groups in each pathway. AD differential signaling pathways within each cell type were identified with a permutation test ($n = 1000$).

## Reporting summary
Further information on research design is available in the Nature Portfolio Reporting Summary linked to this article.

## Data availability
Sample information, gene modules, cluster maker genes, and differentially expressed genes are provided as Supplemental Data files. All bulk and snRNA-seq data are available at Synapse under accession code syn34512705 under controlled use conditions set by human privacy regulations. To access the data, a data use agreement is required that ensures the anonymity of the ROSMAP study participants. Data can be requested at Synapse accession code syn34512705 and is available upon completion of the data use agreement. A data use agreement can be agreed with either SAGE, which maintains Synapse or with Rush University Medical Center through the RADC resource sharing hub (www.radc.rush.edu).

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

## Acknowledgements

The authors are grateful to those who agreed to donate their brains for research. We thank all the employees at RADC for their support

and assistance. This study was supported by NIA grants R01AG074082 and R01AG079223 (to Y.W.), P30AG10161, P30AG72975, R01AG015819, R01AG017917, and U01AG61356 (to D.A.B.), and R01AG061798 (to C.G.). We thank Shinya Tasaki and Bernard Ng for their input on data analysis, Chunjiang Yu for providing A-beta oligomers, Chaeeun Lee and Himanshu Vyas for assisting with culture and RNAscope experiments. We thank Jessica Young and Dirk Keene at the University of Washington for sharing their leptomeninges culture and cryopreservation protocol.

## Author contributions

N.K. and S.D. performed sample preparation and snRNA-seq experiments. J.X. performed snRNA-seq and RNA-seq data preprocessing. C.G. assisted with bulk RNA-seq data analysis. N.K. and A.I. performed computational analysis. N.K. and D.F. performed immunohistochemistry and RNAscope experiments. Z.M. assisted in sample acquisition and derived leptomeningeal cell lines. N.K., A.I., and Y.W. wrote the manuscript. D.A.B. provided ROSMAP resources and support. Y.W. conceived and supervised the study. All authors edited, reviewed, and approved the final manuscript.

## Competing interests

The authors declare no competing interests.
