## [Peer Review File · Nature Communications]

Dissecting the Human Leptomeninges at single-cell resolutionREVIEWER COMMENTS

Reviewer #1 (Remarks to the Author):

In this manuscript by Kerns, Iatrou and colleagues, the authors perform single nuclei transcriptional profiling (snRNAseq) of leptomeninges from healthy and Alzheimer's disease (AD) patients.

This work is significant for several reasons. First, it provides a cellular inventory of the adult aged human leptomeninges, currently data only exists for the human dura and more limited studies of leptomeninges, attached to brain samples taken for snRNAseq. This is a needed resource for the field. Second, it studies a human disease (AD) with meninges involvement (immune cells) and pathology (amyloid plaques) but the role of the meninges in disease progression is understudied, is it causative or bystander? Third, it uses informatic analyses to identify previously unknown cell type specific changes in resident, leptomeningeal border associated macrophages (BAM) and fibroblasts in AD. Recent studies in mice highlight the role of BAMs in meningeal lymphatic function, potentially via interactions with perivascular fibroblasts in perivascular spaces – this study in human should provide disease relevant mechanistic pathways, inferred from their informatic analysis, that could be tested in preclinical animal models as preventative or therapeutic treatments for AD. Finally, the authors integrate their data with an existing human human cerebral parenchymal vasculature snRNAseq data set (Yang et al., 2021) to create a 'map' of the transcriptional diversity of vascular cells moving from the leptomeningeal vasculature to the cerebral vasculature. Given that AD pathology has a significant vascular component and at least one 'path' of exit of waste materials is along the perivascular spaces, transcriptional profiles of the myriad of cell types (endothelial, mural, fibroblast, BAM) along this path is important to understand differential impact of AD pathology (ex: are capillaries vs arterial endothelium differentially impacted in AD?).

Below are critiques related to data presentation and need for additional validation to support some of their claims, organized by moderate and minor comments. Addressing these moderate to minor issues would increase the manuscript's value to the scientific community.

Moderate

1. The samples are all from aged (85+ yo) individuals, raising the possibility that some of the transcriptional profiles and cell inventories are reflective of the advanced age of the donor tissue. Is there any leptomeningeal cell data from brain snRNAseq studies from younger or middle-aged adults? Can the in situ validation be done (or was it performed) on non-aged samples? I do not believe the authors should generate snRNAseq data from non-aged adult samples as this is challenging to obtain, using existing data sets or stating that there is a possibility some of the profiles may reflect what happens in aging (limitation of the study) would be sufficient.

2. Please provide plots show how non-AD vs AD LPM cell classes differ in terms of percent contribution to the clusters (Fig. 1-5) this is important to show if clusters have reduced/more/same representation

from the AD samples. As of now the authors only show the entire data set separated by healthy v AD (Extended Data Figure 2), but not the various subclusters that are presented in the manuscript.

3. Fig. 2: 1) please provide expression plots to show what genes define the mural and endothelial subclusters, 2) not clear what they mean by “arteriovenous axis” in Ln 103-104? Unclear if that statement can be derived from UMAP. 3) vEndothelial_2 cluster is only from the LPM data set, what is the gene expression profile that differentiates this from vEndo_1? 4) 2d: the authors refer to a cluster as ‘venous smooth muscle cell’ however provide no genes that differentiate this cluster from the other two SMC clusters, also validation that these SMCs are only on venous vessels would be important or citations that show that genes enriched in this cluster are only found in venous-associated SMCs (ex: Vanlandewijck et al., 2018 describes a venous SMC in mouse, are similar genes enriched in human?). Please add the addition of the abbreviation meaning of cell clusters in line 106 and 110. Staining for some of these markers along this axis would strengthen and provide validation to the pseudotime analysis.

4. Fig. 3: 1) D, E: the authors provide validation of the genes enriched in the four fibroblast subclusters but it is not clear from their data where these cells are in the leptomeninges layers. Are they near the pial surface or in the upper arachnoid or in the middle, suggesting they are fibroblasts associated with the trabeculae? Or are they perivascular? Adding general spatial information (ex: binning the meninges, showing where each of the 5 cell types are more likely to be located) would provide important information as to where these fibroblast subtypes are located and if these are marking spatially relevant subtypes of fibroblasts (pia, arachnoid or perivascular). They start to do this with the 1b subcluster that is arachnoid barrier cells but doing this for all the clusters would be highly informative for readers. Note: SLC4A4 is high in 1a, this is enriched in arachnoid fibroblasts in mouse (see Desisto et al., 2020 Dev Cell) and Yang et al 2022 appears to show expression in arachnoid trabecula associated fibroblasts in human (though it is hard to tell from the image). Could 1a be arachnoid cells? 2) The explanation and data presentation of how the fibroblast integration with the Yang et al data set is confusing, I think this is in part because there is no validation of where the 4 clusters localize in the leptomeninges, what these different cell types represent. Also, with the data presented, it does not support the claim that perivascular fibroblasts in the LPM vs parenchymal vessels are different. [ex: Ln 237-237; Discussion, Ln 300-301: the authors describe 2a as perivascular and 1a as pial fibroblasts but again, no validation.] 3) 3J: add additional validated arachnoid barrier markers such as Vim, which from the supp data is enriched (see <https://doi.org/10.1084/jem.20220618>).

5. Fig. 4: D: are the different subtypes found in different parts of the leptomeninges (ex: perivascular or in upper/lower or randomly distributed?)

6. Supplemental Figure 1 - bulk transcriptomics of leptomeninges v PFC, this data is not well integrated into the rest of the manuscript, not clear how it is connected to their snRNAseq data or to the study of the LPM cells in general. Later in Ln 217-226, Supplemental Fig 3 the authors talk about the comparison between the normal v AD bulk samples, modules but is this the same sets of data, now split up by normal vs AD? In general how the bulk data is used to support the snRNAseq analysis is more

informative in the context of the AD samples (Supp Fig 3) than how it is used in Supplemental Figure 1.

Minor

Ln 96: leptomeningeal vessels are both in the subarachnoid space and in the pia, please correct this statement.

Ln 150: Sentence appears incomplete. Fibro2a in the frontal cortex and hippocampus are overrepresented?

Signed: Julie Siegenthaler

Reviewer #2 (Remarks to the Author):

This manuscript by Kearns and Iatrou et al. leverages a bioinformatics approach (i.e., single nuclei profiling) to dissect the cellular heterogeneity of the human leptomeninges in a cohort of individuals of advanced aged (~90 years old) and with Alzheimer's disease (AD). Here, the authors were able to detect diverse cell types that comprise the leptomeninges, such as meningeal endothelial, mural, and fibroblast cells. They also identified distinct subtypes of T cells and border associated macrophages (BAMs). Further to this, gene expression profiles revealed a unique transcriptional profile of their BAMs compared to microglia. These BAMs, and fibroblasts, also expressed risk genes for Alzheimer's disease when compared to published GWAS studies. Interestingly, when the authors compared the gene expression profiles of human BAMs vs murine BAMs, they found a high conservation of region-specific gene signatures between the two species. This is an important finding as it will allow the outcomes of studying BAMs in murine models of CNS diseases to have direct relevance to humans.

The authors also made use of in vitro leptomeningeal cultures to further understand the transcriptomic signature of these cellular subtypes. Interestingly, the leptomeningeal cells had a similar signature to that of ex vivo AD fibroblasts when stimulated with amyloid beta oligomers. Lastly, the authors used bioinformatics analyses to explore the ligand-receptor interactions within the leptomeninges to computationally infer intercellular communications. The authors have also used RNA scope and immunohistochemical (IHC) approaches to confirm many of the targets from their single nuclei profiling.

In total, the data provided in the manuscript are novel and of great interest for the field and provide a unique dataset from which the scientific community will benefit. The figures are well put together and the information is clearly presented and described in the text. This is a complete manuscript, and I only have a few minor comments:

Minor comments:

1. If possible, it would be great to identify published single-nuclei data sets that have profiled BAMs and endothelial and mural cells in younger / middle-aged individuals to understand how aging in isolation affects the gene signature of these cells.
2. It would be interesting to compare the gene signatures of the cells from the AD patients to GWAS of other CNS diseases, such as Parkinson's disease and multiple sclerosis, to understand if the risk variants identified are unique to AD or shared across chronic, neurodegenerative diseases.

Reviewer #3 (Remarks to the Author):

The present study has made use of human post-mortem leptomeningeal tissues from individuals with and without diagnosed Alzheimer's disease to determine by bulk and snRNAseq profiling the cellular composition and transcriptional expression profile of leptomeningeal cells. The authors describe differential gene expression profiles in several cells of the leptomeninges of AD patients but mainly in fibroblasts and BAMs. The latter are identified to differentially express several genes identified as risk genes for AD in GWAS studies. Making use of the CellChat algorithm they identify potential cellular communication pathways between the cells identified in the leptomeninges. Exposing cultured leptomeningeal cells to amyloid-beta induces a transcriptional profile resembling that of leptomeningeal fibroblasts in AD patients.

Considering the increasing evidence for a central role of the leptomeninges in shaping CNS immunity this study profiling human leptomeningeal cells is original and of high value. In addition the presented data suggest that the leptomeninges and especially fibroblasts and BAMs contribute to AD pathophysiology.

Although mainly descriptive these data are relevant and provide an excellent reference for animal studies on AD.

A number of issues should still be addressed:

The authors need to better describe how they assigned gene expression profiles to cellular clusters. How did they define what is an endothelial cell, a fibroblasts, an immune cell etc. Endothelial cells are described in Fig. 1c to express PECAM-1, flt1 and IL1R1 – PECAM-1 is however also expressed by immune cells, flt1 is expressed by monocytes and IL1R1 is expressed by numerous cells.

Similarly assignment of the different cellular clusters in the present versus the previous study Yang et al – should be better explained. Expression of which genes identifies capillary, versus arterial and venous endothelial cells?

Arachnoid barrier cells are described to express cld11 and tjp1 – do they also express cdh1 as arachnoid barrier cells do in rodent models?

In Figure 4 the authors describe to perform immunostaining for CD8 to search for T cells. CD8 is only expressed on cytotoxic T cells and on dendritic cells. Immunostaining for CD3 should be performed to search for all T cells in the meninges, this is CD4 and CD8.

Expression meningeal immunity – not so clear as the meninges include the dura mater outside of the CNS as well as the leptomeninges which border the outer CSF filled subarachnoid space and are separated from the dura by the arachnoid barrier – which is a blood-CSF barrier.

In their in vitro culture models the authors describe to grow leptomeningeal fibroblasts. How did they define these cells as fibroblasts? Expression of collagen IV by these cells is puzzling as collagen IV is deposited in endothelial basement membranes. it is not very clear which cell type is

The authors describe ITGA1 and ITGA4 as marker for tissue -resident memory T cells. However, rather ITGAE is the integrin subunit associated with this cell type.

The authors should describe in more depth the tissue isolation protocol. How did they ensure that collection of leptomeninges included all layers of the arachnoid mater in all samples?

In the discussion the authors explain that T cells in the leptomeninges are surrounded by CSF and can thus scan Ags – this is a confusing statement as T cells do not see soluble antigens. Rather the myeloid cells in the CSF would sample Ag and present it to T cells. This should be corrected.

Minor points:

Correct pia matter to pia mater

Correct Figure legend 1

Response to reviewers' comments on NCOMMS-23-20148-T

We thank the reviewers for their appreciation of the novelty and significance of our study and their thoughtful and constructive comments. We addressed them point-by-point and the resulting changes significantly improved our manuscript. Major changes in the revised manuscript are highlighted in red. Below, original comments made by the reviewers are in black, and our responses are in blue. Altogether we added ten new panels (Figures 2G-2I, 3G, 4C,4H, 4I; Extended Figures 1E, 6G, 6J), two figures (Extended Figures 2 and 4), one table (Extended Table 2) and additional tabs to Extended Tables 4 and 9. The revised figure panels are also presented as Response Figures below.

Reviewer #1

In this manuscript by Kerns, Iatrou and colleagues, the authors perform single nuclei transcriptional profiling (snRNAseq) of leptomeninges from healthy and Alzheimer's disease (AD) patients. This work is significant for several reasons. First, it provides a cellular inventory of the adult aged human leptomeninges, currently data only exists for the human dura and more limited studies of leptomeninges, attached to brain samples taken for snRNAseq. This is a needed resource for the field. Second, it studies a human disease (AD) with meninges involvement (immune cells) and pathology (amyloid plaques) but the role of the meninges in disease progression is understudied, is it causative or bystander? Third, it uses informatic analyses to identify previously unknown cell type specific changes in resident, leptomeningeal border associated macrophages (BAM) and fibroblasts in AD. Recent studies in mice highlight the role of BAMs in meningeal lymphatic function, potentially via interactions with perivascular fibroblasts in perivascular spaces – this study in human should provide disease relevant mechanistic pathways, inferred from their informatic analysis, that could be tested in preclinical animal models as preventative or therapeutic treatments for AD. Finally, the authors integrate their data with an existing human cerebral parenchymal vasculature snRNAseq data set (Yang et al., 2021) to create a 'map' of the transcriptional diversity of vascular cells moving from the leptomeningeal vasculature to the cerebral vasculature. Given that AD pathology has a significant vascular component and at least one 'path' of exist of waste materials is along the perivascular spaces, transcriptional profiles of the myriad of cell types (endothelial, mural, fibroblast, BAM) along this path is important to understand differential impact of AD pathology (ex: are capillaries vs arterial endothelium differentially impacted in AD?). Below are critiques related to data presentation and need for additional validation to support some of their claims, organized by moderate and minor comments. Addressing these moderate to minor issues would increase the manuscript's value to the scientific community.

Response: We appreciate the reviewer's precise and thoughtful comments.

1. The samples are all from aged (85+ yo) individuals, raising the possibility that some of the transcriptional profiles and cell inventories are reflective of the advanced age of the donor tissue. Is there any leptomeningeal cell data from brain snRNAseq studies from younger or middle-aged adults? Can the in situ validation be done (or was it performed) on non-aged samples? I do not believe the authors should generate snRNAseq data from non-aged adult samples as this is challenging to obtain, using existing data sets or stating that there is a possibility some of the profiles may reflect what happens in aging (limitation of the study) would be sufficient.

Response: We agree with the reviewer that transcriptional profiles of leptomeningeal cell types may reflect some aging processes. Unfortunately, we are unaware of published snRNAseq datasets from non-aged leptomeninges samples. We agree that comparing gene signatures of meningeal cell types from different age groups would be valuable, and we will have an opportunity to address this issue in future studies. As suggested, we stated this limitation in a new "limitations of the study" section (**lines 366-377**).

2. Please provide plots show how non-AD vs AD LPM cell classes differ in terms of percent contribution to the clusters (Fig. 1-5) this is important to show if clusters have reduced/more/same representation from the AD samples. As of now the authors only show the entire data set separated by healthy v AD (Extended Data Figure 2), but not the various subclusters that are presented in the manuscript.

Response: This is an excellent point. We provided cell-type proportion information for the eight major cell subtypes identified in our study. We performed t-tests between the non-AD (NCI/MCI) and AD groups and did not find differences (pFDR < 0.05) based on our current sample size. We added this data as **Extended Figure 1E** and **new Extended Table 2**. We also conducted similar analyses on the finer cell subtypes identified in Figures 2-5 and did not detect significant differences (**Response Figure 1**). The Fibro_1b subtype showed the most significant proportional differences between the groups but did not pass an FDR significance level.

Response Figure 1. Bar plots (left) and table (right) demonstrate the proportions of finer meningeal cell types between NCI/MCI and AD

3. Fig. 2: 1) please provide expression plots to show what genes define the mural and endothelial subclusters 2) not clear what they mean by “arteriovenous axis” in Ln 103-104? Unclear if that statement can be derived from UMAP. 3) vEndothelial_2 cluster is only from the LPM data set, what is the gene expression profile that differentiates this from vEndo_1? 4) 2d: the authors refer to a cluster as ‘venous smooth muscle cell’ however provide no genes that differentiate this cluster from the other two SMC clusters, also validation that these SMCs are only on venous vessels would be important or citations that show that genes enriched in this cluster are only found in venous-associated SMCs (ex: Vanlandewijck et al., 2018 describes a venous SMC in mouse, are similar genes enriched in human?). Please add the addition of the abbreviation meaning of cell clusters in line 106 and 110. Staining for some of these markers along this axis would strengthen and provide validation to the pseudotime analysis.

Response: Thank you for your suggestions. We added **Figure 2i** and **Extended Figure 2** to include 1) violin plots for the expression of top marker genes for endothelial and mural clusters; 2) volcano plots to highlight the differential expressed genes between the arterial and venous endothelial cells and between the arterial and venous mural cells; 3) volcano plots to highlight the differential expressed genes between vEndo1 and vEndo2 and 4) volcano plots to highlight the differential expressed genes between aSMC1 and aSMC2. We edited the text (**lines 101-103; 107-110**) and updated **Extended Table 4** with the complete differential profiles for these contrasts. The added figure panels are also presented below (**Response Figure 2a-2f**).

As suggested, we compared the venous SMCs to mouse SMCs by leveraging the single-cell RNAseq data from Muhl et al.¹. As reported by Muhl et al., we found that both vSMC and aSMC clusters express ACTA2, but only the aSMC cluster expresses CSPG4. We validated this result by IHC, showing ACTA2+CSPG4+ arterial SMCs and ACTA2+CSPG4- venous SMCs. In addition, we examined the expression of a mouse vSMC gene module (a set of mouse vSMC marker genes from Muhl et al.) in the leptomenigeal SMCs and

detected its enrichment only in the vSMC cluster (**Figure 2g-2h**). We edited the text (**lines 112-122**). Those added panels are also presented below (**Response Figure 2g-2h**).

The “arteriovenous axis in Ln103-104 referred to the anatomical arteriovenous from leptomeningeal arteries to parenchymal arterioles, capillaries, venules, and back to leptomeningeal veins. We have removed this sentence to avoid confusion. We added the missing abbreviations in the text (**lines 97, 105**).

Response Figure 2. Detailed transcriptomic profiles of the endothelial and mural cell subtypes. (a) Violin plot of the top 5 differentially expressed genes between endothelial subtypes (b-c) Volcano plot of the differentially expressed genes distinguishing venous from arterial endothelial cells (b), and the two venous endothelial cell subtypes (c) with a threshold set at $|\log FC| > 0.5$ and $p_{BON} < 0.01$. (d) Violin plot of the top 5 differentially expressed genes between mural subtypes (e-f) Volcano plot of the differentially expressed genes distinguishing venous from arterial smooth muscle cells (b) and two arterial smooth muscle cell subtypes with a threshold set at $|\log FC| > 0.5$ and $p_{BON} < 0.01$. (g) Violin plots of ACTA2 and CSPG4 expression across the smooth muscle cell types (SMC) identified in the current study and representative chromogenic immunohistochemistry. Scale bars = 100 μ m. a= artery, v= vein. (h) Enrichment of mouse venous SMC gene markers (Muhl et al., 2022) in the leptomeningeal SMC cells.

4. Fig. 3: 1) D, E: the authors provide validation of the genes enriched in the four fibroblast subclusters but it is not clear from their data where these cells are in the leptomeninges layers. Are they near the pial surface or in the upper arachnoid or in the middle, suggesting they are fibroblasts associated with the trabeculae? Or are they perivascular? Adding general spatial information (ex: binning the meninges, showing where each of the 5 cell types are more likely to be located) would provide important information as to where these fibroblast subtypes are located and if these are marking spatially relevant subtypes of fibroblasts (pia, arachnoid or perivascular). They start to do this with the 1b subcluster that is arachnoid barrier cells but doing this for all the clusters would be highly informative for readers. Note: SLC4A4 is high in 1a, this is enriched in arachnoid fibroblasts in mouse (see Desisto et al., 2020 Dev Cell) and Yang et al 2022 appears to show expression in arachnoid trabecula associated fibroblasts in human (though it is hard to tell from the image). Could 1a be arachnoid cells?

Response: We agree with the reviewer that mapping fibroblast subtypes' spatial location(s) would be valuable for understanding their cellular functions. To this end, we reanalyzed our RNAscope data on SLC7A2 (Fibro_1a and Fibro_1b), LAMA2 (Fibro_2a and Fibro_2b), and TPRM3 (Fibro_1b and Fibro_2a). As suggested, we binned cells based on proximity to large and small vessels (large vessels > 100 μ m; small vessel < 100 μ m) and subarachnoid space but did not detect specific spatial patterns of labeled fibroblast subtypes (**Response Figure 3a**). Our IHC staining of SLC4A4 (Fibro_1a), LEPR (Fibro_1b), and CEMIP (Fibro_1b, 2a, 2b) on postmortem leptomeninges also did not reveal apparent spatial pattern (**Response Figure 3b**). We admit that those data are preliminary due to limited workable antibodies and probes. Future spatial transcriptomic studies will provide more precise spatial characterizations of leptomeningeal cell types. With this caveat, we only named the fibroblast subtypes by their transcriptional profiles, i.e.,

Fibro_1a, 1b, 2a, 2b, in our revised manuscript (**lines 150-156**). We also noted this caveat in the “limitations of the study” section (**lines 366-377**).

The explanation and data presentation of how the fibroblast integration with the Yang et al data set is confusing, I think this is in part because there is no validation of where the 4 clusters localize in the leptomeninges, what these different cell types represent. Also, with the data presented, it does not support the claim that perivascular fibroblasts in the LPM vs parenchymal vessels are different. [ex: Ln 237-237; Discussion, Ln 300-301: the authors describe 2a as perivascular and 1a as pial fibroblasts but again, no validation.]

Response: We appreciate these comments. We agree that transferring the cell-type annotations from the Yang et al. study based on integrated transcriptomic data needs to be clarified. Indeed, the annotations of brain fibroblasts are controversial in the field due to conflicting conclusions from recent studies on whether the *SLC4A4*+ *KCNMA1*+ fibroblasts are meningeal and whether any or all the others are perivascular^{2,3}. In addition, our preliminary data (**Response Figure 3**) does not support a specific spatial pattern of any subtype. Therefore, we removed claims of “perivascular” or “meningeal” fibroblast types in our revision and only named them by their transcriptional signature. We also moved the integration data from Figure 3 to **Extended Figure 3** and edited the text accordingly (**lines 150-156**).

3J: add additional validated arachnoid barrier markers such as Vim, which from the supp data is enriched (see <https://doi.org/10.1084/jem.20220618>).

Response: This is a great suggestion. Of the four markers in the suggested paper, VIM and CDH1 are expressed and enriched in Fibro_1b. We added this result to **Figure 3G** and edited the text accordingly (**lines 150-156**). MUC1 and PGR are detectable in Fibro_1a and Fibro_1b subtypes but with a shallow detection rate. We included those four genes and our initially selected genes in the dot plot below (**Response Figure 4**).

Response Figure 4. The dot plot demonstrates the average expression and detection rate of arachnoid barrier markers among four fibroblast subtypes.

5. Fig. 4: D: are the different subtypes found in different parts of the leptomeninges (ex: perivascular or in upper/lower or randomly distributed?)

Response: This is a great question. Our IHC staining for T and BAM cell markers shows those cells are distributed throughout the leptomeninges (**Figure 4c, 4h, 4j**) without enrichment around vessels or in the upper or lower part of the leptomeninges. However, our analysis was primarily done on a limited number of samples with a limited set of workable probes and antibodies. Therefore, we cannot exclude the possibility that particular subtypes of immune cells may present delicate spatial patterns. This warrants future spatial transcriptomics studies to map the spatial location of leptomeningeal cell types. We have stated this in the “limitations of the study” section (**lines 366-377**).

6. Supplemental Figure 1 - bulk transcriptomics of leptomeninges v PFC, this data is not well integrated into the rest of the manuscript, not clear how it is connected to their snRNAseq data or to the study of the LPM cells in general. Later in Ln 217-226, Supplemental Fig 3 the authors talk about the comparison between the normal v AD bulk samples, modules but is this the same sets of data, now split up by normal vs AD? In general how the bulk data is used to support the snRNAseq analysis is more informative in the context of the AD samples (Supp Fig 3) than how it is used in Supplemental Figure 1.

Response: We agree with the reviewer’s comment. We consolidated bulk RNAseq data into one new figure to support AD molecular phenotype (**Extended Figure 5**). The bulk RNAseq was performed on leptomeninges from 44 individuals (21 NCI/MCI and 23 AD cases). We then constructed gene co-expression modules for the bulk RNAseq data and examined their preservation against parenchymal (dorsolateral prefrontal cortex) gene modules⁴. We identified shared (preserved) and leptomeninges-specific (non-preserved) modules and then performed a module-trait analysis to associate modules with AD-related traits. Module 41 was the only module associated with Alzheimer’s dementia after multiple testing corrections (FDR≤0.05). This gene module is enriched in the extracellular matrix and immune function, in line with our snRNAseq results. We edited the text (**lines 221-237**) and presented this figure below (**Response Figure 5**).

Response Figure 5. Leptomeningeal gene co-expression modules and their association with AD traits.

(a) Schematic illustration of experimental design, bulk RNA-seq, and module-trait association analysis. (b) Co-expressed gene modules generated using SpeakEasy and their gene set size and preservation scores in the DLPFC data from Mostafavi et al., 2019. Modules with a preservation score (z.summary) lower than 7.5 are considered non-preserved. (c) Association between modules and AD-related traits, including clinical and pathological diagnosis, cerebral amyloid angiopathy (CAA), cognitive decline, plaques, and tangles; * $p \leq 0.05$, ** $p_{FDR} \leq 0.05$. (d) Pathway analysis of the gene members of module 41. Significant Reactome pathways are visualized ($p_{BH} < 0.05$).

7. Ln 96: leptomeningeal vessels are both in the subarachnoid space and in the pia, please correct this statement. Ln 150: Sentence appears incomplete. Fibro2a in the frontal cortex and hippocampus are overrepresented?

Response: We removed Ln150 and edited Ln 96 (line 91).

Reviewer #2

This manuscript by Kearns and Iatrou et al. leverages a bioinformatics approach (i.e., single nuclei profiling) to dissect the cellular heterogeneity of the human leptomeninges in a cohort of individuals of advanced aged (~90 years old) and with Alzheimer's disease (AD). Here, the authors were able to detect diverse cell types that comprise the leptomeninges, such as meningeal endothelial, mural, and fibroblast cells. They also identified distinct subtypes of T cells and border associated macrophages (BAMs). Further to this, gene expression profiles revealed a unique transcriptional profile of their BAMs compared to microglia. These BAMs, and fibroblasts, also expressed risk genes for Alzheimer's disease when compared to published GWAS studies. Interestingly, when the authors compared the gene expression profiles of human BAMs vs murine BAMs, they found a high conservation of region-specific gene signatures between the two species. This is an important finding as it will allow the outcomes of studying BAMs in murine models of CNS diseases to have direct relevance to humans. The authors also made use of in vitro leptomeningeal cultures to further understand the transcriptomic signature of these cellular subtypes. Interestingly, the leptomeningeal cells had a similar signature to that of ex vivo AD fibroblasts when stimulated with amyloid beta oligomers. Lastly, the authors used bioinformatics analyses to explore the ligand-receptor interactions within the leptomeninges to computationally infer intercellular communications. The authors have also used

RNA scope and immunohistochemical (IHC) approaches to confirm many of the targets from their single nuclei profiling. In total, the data provided in the manuscript are novel and of great interest for the field and provide a unique dataset from which the scientific community will benefit. The figures are well put together and the information is clearly presented and described in the text. This is a complete manuscript, and I only have a few minor comments:

Response: We appreciate the reviewer's thoughtful and encouraging comments.

1. If possible, it would be great to identify published single-nuclei data sets that have profiled BAMs and endothelial and mural cells in younger / middle-aged individuals to understand how aging in isolation affects the gene signature of these cells.

Response: : We agree with the reviewer. Reviewer 1 made the same comment. See **Response to Reviewer 1.1**.

2. It would be interesting to compare the gene signatures of the cells from the AD patients to GWAS of other CNS diseases, such as Parkinson's disease and multiple sclerosis, to understand if the risk variants identified are unique to AD or shared across chronic, neurodegenerative diseases.

Response: This is an excellent point. We extended our GWAS gene cell-type enrichment analysis to other neurodegenerative disorders, including Parkinson's Disease (PD), Multiple Sclerosis (MS), and Frontotemporal Dementia (FTD) (**Extended Figure 4 and Extended Table 9**). We compiled disease-specific GWAS genes from the latest studies⁵⁻⁷ and assessed their cell-type enrichment in each disease as we did for AD. Interestingly, we observed enrichment of PD GWAS genes in monocytes and pericytes, MS GWAS genes in all immune cell types, aEndo and capillary Endo cells, but no cell-type enrichments for FTD. We added the results to the revised manuscript (**lines 205-211**) and presented this result below (**Response Figure 6**).

Response Figure 6. Disease-specific GWAS gene expression in leptomenigeal cell types. (a-c) Heatmap of GWAS gene's proportional expression in each detected cell type across Parkinson's disease (a), Multiple sclerosis (b), and frontotemporal dementia (c). The values on the bottom represent the relative expression of all GWAS genes in each cell type. * denotes cell types with enriched expression for the respective GWAS genes.

Reviewer #3

The present study has made use of human postmortem leptomenigeal tissues from individuals with and without diagnosed Alzheimer's disease to determine by bulk and snRNAseq profiling the cellular composition and transcriptional expression profile of leptomenigeal cells. The authors describe differential gene expression profiles in several cells of the leptomeninges of AD patients but mainly in fibroblasts and

BAMs. The latter are identified to differentially express several genes identified as risk genes for AD in GWAS studies. Making use of the CellChat algorithm they identify potential cellular communication pathways between the cells identified in the leptomeninges. Exposing cultured leptomeningeal cells to amyloid-beta induces a transcriptional profile resembling that of leptomeningeal fibroblasts in AD patients. Considering the increasing evidence for a central role of the leptomeninges in shaping CNS immunity this study profiling human leptomeningeal cells is original and of high value. In addition the presented data suggest that the leptomeninges and especially fibroblasts and BAMs contribute to AD pathophysiology. Although mainly descriptive these data are relevant and provide an excellent reference for animal studies on AD.

Response: We really appreciate the reviewer's encouraging comments.

1. The authors need to better describe how they assigned gene expression profiles to cellular clusters. How did they define what is an endothelial cell, a fibroblast, an immune cell etc. Endothelial cells are described in Fig. 1c to express PECAM-1, flt1 and IL1R1 – PECAM-1 is however also expressed by immune cells, flt1 is expressed by monocytes and IL1R1 is expressed by numerous cells.

Response: This is an important suggestion. Because the human leptomeninges has not been previously profiled, we primarily relied on reported classical cell-type marker genes for annotation, especially the recently published snRNAseq datasets from brain vasculature and human dura^{2,3,8}. It is worth noting that most genes are not uniquely expressed in one cell type. Therefore, we confirmed the expression of multiple marker genes in a particular cell cluster to ensure accurate annotations. Specifically, the endothelial clusters were identified by the co-expression of PECAM1, CLND5, FLT1, mural cells by co-expression of ACTA2, MYH11, CSPG4, and COL4A1, fibroblasts by co-expression of DCN, with SLC4A4 or LAMA2, and immune cells by co-expression of PTPRC with either MRC1 and F13A1 (immune_1) or PARP8 and THEMIS (immune_2). All clusters showed an absence or significantly lower expression of the key markers of other cell types. For example, endothelial, mural, and fibroblast clusters lacked PTPRC expression (**Figure 1c**). When appropriate antibodies were available, we performed IHC to validate the expression of key cell-type markers (**Figure 1d**). For subclusters where we identified transcriptionally distinct cell types but did not have appropriate datasets to cross-reference, we annotated these clusters numerically, e.g., aSMC_1, aSMC_2, etc. For fibroblast and BAM subtypes, we also used RNAscope to confirm the transcriptionally distinct cell populations in the postmortem leptomeningeal tissue. We have added a more detailed description to the **Methods** section (**lines 487-494**).

2. Similarly assignment of the different cellular clusters in the present versus the previous study Yang et al – should be better explained. Expression of which genes identifies capillary, versus arterial and venous endothelial cells?

Response: We thank the reviewer for the comments. The assignment of cells to the clusters was based on transcriptional similarity to the Yang et al. cell-type clusters. First, we integrated the two datasets and then transferred the Yang et al. annotations to the transcriptionally similar leptomeningeal cells. We detected three major endothelial cell clusters (aEndo_1, vEndo_1, and vEndo_2), two arterial/arteriole smooth muscle cell clusters (aSMC_1 and aSMC_2), and a unique venous smooth muscle cell cluster (vSMC) cluster. As suggested, we added **Figure 2i** and **Extended Figure 2** to include 1) violin plots for the expression of top marker genes for endothelial and mural clusters; 2) volcano plots to highlight the differential expressed genes between the arterial and venous endothelial cells and between the arterial and venous mural cells; 3) volcano plots to highlight the differential expressed genes between vEndo1 and vEndo2 and 4) volcano plots to highlight the differential expressed genes between aSMC1 and aSMC2. We also updated **Extended Table 4** with the complete differential profiles for these contrasts. In addition, we validated the vSMC identity (**Figure 2g-2h**) by leveraging Muhl et al. single-cell RNAseq data of mouse SMCs¹. We edited the text (**lines 101-122**). Since capillary endo cells were rare in our leptomeningeal dataset but were extensively characterized by Yang et al., we did not describe them further. We added a

more detailed description to the **Methods** section (lines 487-494). The added figure panels are also presented in **Response Figure 1**.

3. Arachnoid barrier cells are described to express *cld11* and *tjp1* – do they also express *cdh1* as arachnoid barrier cells do in rodent models?

Response: We checked the *cdh1* expression in fibroblast subtypes. The Fibro_1b arachnoid barrier cells express CDH1 transcripts despite a low detection rate in our snRNAseq data. We expanded the arachnoid barrier marker gene panel to include CDH1 and VIM in **Figure 3G**; also see **Response Figure 4**.

4. In Figure 4 the authors describe to perform immunostaining for CD8 to search for T cells. CD8 is only expressed on cytotoxic T cells and on dendritic cells. Immunostaining for CD3 should be performed to search for all T cells in the meninges, this is CD4 and CD8.

Response: We conducted IHC using CD3 antibody and updated Figure 4c. We also stained the adjacent sections with F13A1 antibody and updated Figure 4h.

5. Expression meningeal immunity – not so clear as the meninges include the dura mater outside of the CNS as well as the leptomeninges which border the outer CSF filled subarachnoid space and are separated from the dura by the arachnoid barrier – which is a blood-CSF barrier.

Response: We added a paragraph in the introduction to discuss the structure and barriers of meninges (lines 35-47).

6. In their in vitro culture models the authors describe to grow leptomeningeal fibroblasts. How did they define these cells as fibroblasts? Expression of collagen IV by these cells is puzzling as collagen IV is deposited in endothelial basement membranes. it is not very clear which cell type is

Response: We agree that the markers used, such as fibroblast specific antigen and collagen IV, also label other cell types. To investigate this further, we mapped the gene signature of ex vivo cell types to cultured leptomeningeal cells and found enrichment of mural and fibroblast cell types (**Response Figure 7A**). Interestingly, when we mapped the gene signature of ex vivo cell-type genes and AD differential expressed genes to the A β -induced genes of cultured cells, we only detected an enrichment of fibroblasts but not mural cells (**Response Figure 7B and 7C**). Therefore, cultured leptomeningeal cells behave more like fibroblasts upon A β treatment. However, due to the mixed signature of those cells at the baseline, we replaced the term "fibroblast-like cells" with "leptomeningeal cultures" in our revised manuscript (line 255).

Response Figure 7. Mapping the gene signature of ex vivo cell types to cultured leptomeningeal cells. A) Geneset enrichment of the ex vivo cell cluster gene markers ($p < 0.01$ and $\log FC > 1$) in the gene signature of cultured leptomeningeal cells B) Geneset enrichment of ex vivo cell clusters in the pre-ranked list of the in vitro A β -induced DEGs C) Enrichment score of the ex vivo fibroblast AD DEGs (blue) and ex vivo mural AD DEGs (red) in the pre-ranked list of the in vitro A β -induced DEGs.

7. The authors describe ITGA1 and ITGA4 as marker for tissue -resident memory T cells. However, rather ITGAE is the integrin subunit associated with this cell type.

Response: We edited the text accordingly.

8. The authors should describe in more depth the tissue isolation protocol. How did they ensure that collection of leptomeninges included all layers of the arachnoid mater in all samples?

Response: We added a more detailed description in the **methods** section (**lines 405-416**). All autopsies in this study were performed at the Rush Alzheimer’s Disease Center following a PMI of ~8.5 hours. We adopted a published protocol for leptomeninges dissection and cryo-preservation⁹. Briefly, the skull with the adhered dura was first removed, then the brainstem was severed at the level of the mammillary bodies. The cerebrum with attached leptomeninges was hemisected to ensure that the leptomeninges were intact, as assessed by the presence of blood vessels and arachnoid trabeculae. The leptomeninges were then stripped from the motor cortex region using forceps, placed in ice-cold collection media, and transferred to the lab on ice for further processing. To avoid sampling bias, we further examined the sample representation in each cell type following data harmonization and clustering analyses. We found that most cell clusters contained cells from 80-100% of the participants except for two rare cell types, B cells and Tcell_2 (**Response Figure 8**). Therefore, our standard dissection methods produced minimal sampling bias between study participants. Having stated this, we cannot exclude the possibility that rare cell types are missed in our study due to our limited sample size, as detailed in the “limitations of the study” section (**lines 366-377**).

Response Figure 8. Bar chart of the percent of participants (n=18) represented by each leptomeningeal cell type. Cell clusters are ordered from largest to smallest.

9. In the discussion the authors explain that T cells in the leptomeninges are surrounded by CSF and can thus scan Ags – this is a confusing statement as T cells do not see soluble antigens. Rather the myeloid cells in the CSF would sample Ag and present it to T cells. This should be corrected.

Response: We edited the text (**lines 320-321**).

10. Minor points: Correct pia matter to pia mater; Correct Figure legend 1

Response: We edited the text.

References

1. Muhl, L. *et al.* A single-cell transcriptomic inventory of murine smooth muscle cells. *Dev. Cell* **57**, 2426-2443.e6 (2022).
2. Yang, A. C. *et al.* A human brain vascular atlas reveals diverse mediators of Alzheimer's risk. *Nature* **603**, 885–892 (2022).
3. Garcia, F. J. *et al.* Single-cell dissection of the human brain vasculature. *Nature* **603**, 893–899 (2022).
4. Mostafavi, S. *et al.* A molecular network of the aging human brain provides insights into the pathology and cognitive decline of Alzheimer's disease. *Nat. Neurosci.* **21**, 811–819 (2018).
5. Chang, D. *et al.* A meta-analysis of genome-wide association studies identifies 17 new Parkinson's disease risk loci. *Nat. Genet.* **49**, 1511–1516 (2017).
6. INTERNATIONAL MULTIPLE SCLEROSIS GENETICS CONSORTIUM. Multiple sclerosis genomic map implicates peripheral immune cells and microglia in susceptibility. *Science* **365**, eaav7188 (2019).
7. Pottier, C. *et al.* Potential genetic modifiers of disease risk and age at onset in patients with frontotemporal lobar degeneration and GRN mutations: a genome-wide association study. *Lancet Neurol.* **17**, 548–558 (2018).
8. Wang, A. Z. *et al.* Single-cell profiling of human dura and meningioma reveals cellular meningeal landscape and insights into meningioma immune response. *Genome Med.* **14**, 1–25 (2022).
9. Rose, S. E. *et al.* Leptomeninges-Derived Induced Pluripotent Stem Cells and Directly Converted Neurons From Autopsy Cases With Varying Neuropathologic Backgrounds. *J. Neuropathol. Exp. Neurol.* **77**, 353–360 (2018).

REVIEWERS' COMMENTS

Reviewer #1 (Remarks to the Author):

The authors have done an excellent job revising the manuscript in response to my extensive comments, with new/revised analysis and edits throughout the manuscript. I appreciate their attempts to address my comments about the spatial location of fibroblast clusters subtypes and agree that current methods (with just a few genes/proteins) it is challenging to make any firm conclusions about fibroblast heterogeneity, future work using different methods (spatial transcriptomics) should prove useful to addressing this point. In its current form, I believe this work is a valuable resource for the field of meninges biology and support its publication.

Reviewer #2 (Remarks to the Author):

I would like to thank the authors for their responses to my comments. The addition of the GWAS gene cell-type enrichment analysis to the other neurodegenerative disorders was very interesting and informative. After having read the other reviewers' comments, the associated author responses, and the revised manuscript there has been an overall improvement since the original submission. I have only two minor comments to make on the revised manuscript, though these should not hold up the acceptance and eventual publication of the manuscript.

Minor comments:

1. Between lines 223 and 224, the authors provide a breakdown of the patients. However, there is a small typo in the text. It reads "...(AD=23, NCI/MCI=2; Table S1; Extended Fig. 5a) when it should be "...(AD=23, NCI/MCI=21; Table S1; Extended Fig. 5a) as the total number of patients used for sequencing equaled 44.
2. I would suggest the authors indicate the sequencing (both bulk and single nuclei) was performed on aged or super-aged individuals whenever they reference its novelty related to the first comprehensive sequencing of the human leptomeninges. None of the patients profiled were younger than 90 years old, therefore it is misleading to claim these data come from a wide spectrum of human ages. One such claim is provided as an example:

"However, no single-nuclei RNA-seq study on dissected human leptomeninges has been conducted to date. Therefore, there is a critical need to unbiasedly chart cell types and states of human leptomeninges in normal aging and disease. We address this challenge by reporting the first comprehensive single-nuclei characterization of 42,557 cells from isolated postmortem human leptomeninges. Our study provides a transcriptomic atlas of the human leptomeninges, revealing rich cell type diversity within the stromal and immune cell types".

Reviewer #3 (Remarks to the Author):

I thank the authors for their thoroughness in addressing all queries.

Manuscript NCOMMS-23-04988A
Response to the Reviewer #2

We thank the Editor and Reviewers for their appreciation of the significant improvement of the revised manuscript and their support of its suitability for publication in *Nature Communications*. There were no further queries from Reviewers #1 and #3. Two minor comments from Reviewer #2 are addressed below. We also addressed queries from the Author Checklist and Reporting Summary; the edits are in red in the revised manuscript.

1. Between lines 223 and 224, the authors provide a breakdown of the patients. However, there is a small typo in the text.

Response: We corrected this typo, highlighted in red.

2. I would suggest the authors indicate the sequencing (both bulk and single nuclei) was performed on aged or super-aged individuals whenever they reference its novelty related to the first comprehensive sequencing of the human leptomeninges.

Response: We added this clarification to lines 62, 87, 305, and 339, highlighted in red.